# XPL: A Cross-Model framework for Semi-Supervised Prompt Learning in Vision-Language Models

**Omprakash Chakraborty**  *opckgp@iitkgp.ac.in*
*IIT Kharagpur*

**Aadarsh Sahoo**  *aadarsh@ibm.com*
*MIT-IBM Watson AI Lab*

**Rameswar Panda**  *rpanda@ibm.com*
*MIT-IBM Watson AI Lab*

**Abir Das**  *abir@cse.iitkgp.ac.in*
*IIT Kharagpur*

**Reviewed on OpenReview:** *https://openreview.net/forum?id=oxAZv3QD6M*

## Abstract

Prompt learning, which focuses on learning soft prompts, has emerged as a promising approach for efficiently adapting pretrained vision-language models (VLMs) to multiple downstream tasks. While prior works have shown promising performances on common benchmarks, they typically rely on labeled data samples only. This greatly discredits the information gain from the vast collection of otherwise unlabeled samples available in the wild. To mitigate this, we propose a simple yet efficient cross-model framework to leverage on the unlabeled samples achieving significant gain in model performance. Specifically, we employ a semi-supervised prompt learning approach which makes the learned prompts invariant to the different views of a given unlabeled sample. The multiple views are obtained using different augmentations on the images as well as by varying the lengths of visual and text prompts attached to these samples. Experimenting with this simple yet surprisingly effective approach over a large number of benchmark datasets, we observe a considerable improvement in the quality of soft prompts thereby making an immense gain in image classification performance. Interestingly, our approach also benefits from out-of-domain unlabeled images highlighting the robustness and generalization capabilities. [1]

## 1 Introduction

Recently vision-language models (VLMs) (Jia et al., 2021; Li et al., 2022; 2021; Radford et al., 2021; Wu et al., 2021; Fini et al., 2023; Jiang et al., 2023) have shown encouraging progress on a number of downstream tasks. These models are initially trained on large-scale data to align language and vision modalities. Such a paradigm allows zero-shot transfer to downstream tasks since one can synthesize a natural language description known as *prompt* of the new class (*e.g.*, 'a photo of a class name') to be fed to the text encoder and compare the generated text features with visual features. However, the non-trivial task of choosing the best hand-crafted prompts is difficult, requiring a lot of time and domain-specific heuristics. This has led to prompt learning (Lu et al., 2022; Zhou et al., 2022b;a; Wang et al., 2023). It aims to use soft prompts that are learned using labeled samples from downstream tasks, keeping the pretrained model frozen. These approaches have demonstrated comparable performance to full fine-tuning though learning only few parameters and are known to adapt to new tasks quickly (He et al., 2022).

---

[1] Project page: `https://cvir.github.io/projects/xpl`

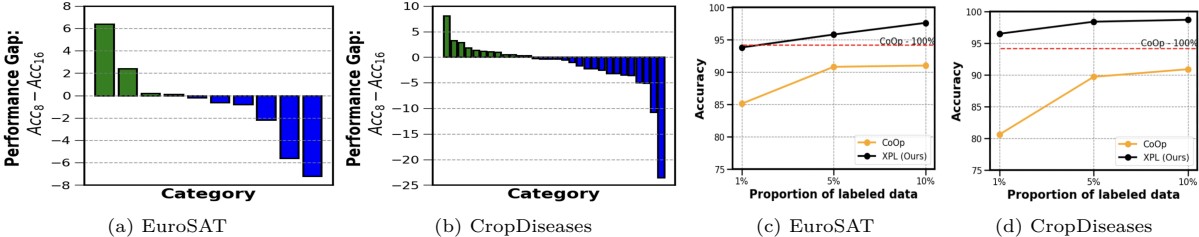

| (a) EuroSAT | (b) CropDiseases | (c) EuroSAT | (d) CropDiseases |

Figure 1: **(a, b)**: Category-wise performance gap between two models leveraging same amount of labeled and unlabeled data but with different number of learnable prompts (8 and 16 textual and visual prompts) on EuroSAT and CropDiseases respectively. $Acc_8$ and $Acc_{16}$ denote the accuracy with 8 and 16 length prompts respectively showing the complimentary knowledge acquired by the two models. **(c, d)**: comparison of `XPL` with the conventional text-only prompt learning CoOp (Zhou et al., 2022b) trained using different percentages of labeled training data on the same datasets. With only 1% of labeled data, `XPL` surpasses the fully supervised CoOp (shown with red dotted line). CoOp with same amount of labeled data fail to reach the accuracy of `XPL`.

To the best of our knowledge, prompt learning has thus far relied only on supervised approaches, which makes it critically dependent on heavily curated data requiring tedious human labeling effort. This motivates us to look beyond traditional supervised prompt learning in order to not only minimize the annotation effort but also to improve the performance on downstream tasks in extremely low labeled data regime. Semi-supervised Learning (SSL) has shown promising results in visual scene understanding. Among these, self-training or pseudolabeling (Arazo et al., 2020) uses confident predictions of unlabeled samples as true label for further training. Consistency regularization (Bachman et al., 2014) transforms unlabeled samples to different views and forces the model to learn invariant representations. However, in low-labeled data regime, the learned representations tend to lack enough discriminative power for downstream tasks. To handle this issue, works like (Xu et al., 2022) employs not single, but multiple models towards cross-model representation learning leveraging the complementary representations from these different models. Although these approaches have shown promising results, the strength of semi-supervised learning has not been harnessed in prompt learning for large VLMs. In this work, we show that semi-supervised prompt learning not only exploits the unlabeled data present in hand but also helps learn richer representations without additional manual labeling.

While prompt learning is an efficient and quick adaptation paradigm, their low capacity may not allow a single prompt learning model to achieve best performances in all. To better exploit multiple prompt learners, we present a semi-supervised approach based on the complementary representations at the model level. We observe that two models leveraging unlabeled data but with different number of learnable prompts exhibit markedly different category-wise performance (ref. Figure 1a and b). This indicates that the two models learn complimentary knowledge and thus can complement in providing semi-supervision to each other. To this end, we introduce our semi-supervised **C**ross-model **P**rompt **L**earning (`XPL`) approach that relies on the invariance of the learned prompts to different views of unlabeled data. Given a pretrained VLM, we create a set of augmented versions of the unlabeled data and pass them via two pathways (known as the *primary* and the *auxiliary* pathways) each having a different length of soft prompts associated to them. Then, given an unlabeled image, we bring a confident prediction from the auxiliary network as the pseudo-label for the primary and vice versa facilitating a greater engagement of unlabeled images. To the best of our knowledge, `XPL` is one of the first works in semi-supervised prompt learning in VLMs. We evaluate our approach on different image classification tasks in 15 standard datasets from diverse categories including Aerial, Medical, Natural, Illustrative, Texture, Symbolic and Structured images. We focus on learning prompts at significantly low labeled data regime, which includes the conventional few-shot classification settings as well as settings involving various proportions of labeled training data. Figure 1c and d show that using only 1% training data with labels and rest as unlabeled data, `XPL` superseeds the performance of the supervised text-only prompt learning approach CoOp (Zhou et al., 2022b) that uses 100% training data with labels in the benchmark datasets of EuroSAT (Helber et al., 2019) and CropDiseases (Mohanty et al., 2016) respectively. `XPL` is also shown to be consistently better than CoOp that uses the same amount of labeled data as ours showing the advantage of multimodal, semi-supervised and cross-model approach for prompt learning.

## 2 Related Works

**Vision Language Models (VLMs).** Development of VLMs employing single-stream (Su et al., 2019; Chen et al., 2020b; Li et al., 2019; 2020) or dual-stream (Tan & Bansal, 2019; Goel et al., 2022; Jia et al., 2021; Li et al., 2022; 2021; Radford et al., 2021; Jiang et al., 2023) paradigms have progressed significantly. The prevailing dual-stream paradigm, which separates the image encoder and text encoder, forms the backbone of our approach. By enabling zero-shot transfer to a range of downstream tasks, works like CLIP (Radford et al., 2021) and ALIGN (Jia et al., 2021) have substantially changed computer vision lately. Few methods have learned transferable features using additional supervision (Li et al., 2021; Mu et al., 2021), finer-grained interactions (Yao et al., 2021), modern Hopfield networks (Fürst et al., 2021), optimal transport distillation (Wu et al., 2021), cycle consistency (Goel et al., 2022), and hierarchical feature alignment (Gao et al., 2022). However, these use supervised training only. Ours is one of the first works that goes beyond the supervised setting and learns prompts leveraging on unlabeled data alongside a few labeled samples.

**Prompt Learning.** There have been numerous studies on prompt tuning (Huang et al., 2022; Zhou et al., 2022b) for effective adaption of VLMs. CoOp (Zhou et al., 2022b), a well-known prompt tuning framework draws its inspiration from NLP (Lester et al., 2021; Zhong et al., 2021; Zhou et al., 2022c; Sordoni et al., 2024) and uses cross-entropy loss to learn prompt vectors. UPL (Huang et al., 2022) proposes an unsupervised prompt learning framework without necessitating any annotations of the target dataset, while, ProDA (Lu et al., 2022) learns various prompts from data to manage the variation of visual representations. Some approaches like CLIP-Adapter (Gao et al., 2021) and Tip-Adapter (Zhang et al., 2021) adjust VLMs by training additional adapter networks using labeled data. In (Shu et al., 2022), a framework for test-time prompt tuning is also proposed that does not require training data or annotations. These methods outperform hand-crafted prompts in a reasonable variety of ways, but they frequently have low generalizability when there are changes in the data distribution.

**Semi-Supervised Learning.** Semi-supervised learning (SSL) comprises of several techniques (Chapelle et al., 2009) to utilize unlabeled data for considerably reducing the dependency on annotations. Many efficient approaches have been proposed over time. For instance, self-training with pseudo-labels (Arazo et al., 2020; Grandvalet & Bengio, 2005; Lee, 2013), contrastive learning (Singh et al., 2021) and consistency regularization (Bachman et al., 2014; Berthelot et al., 2019a;b; Miyato et al., 2018) have shown to significantly enhance the performance over their supervised counterparts. Another current trend for SSL is the use of self-supervised learning techniques like rotation prediction (Gidaris et al., 2018), discriminative image transformation (Dosovitskiy et al., 2014) *etc.* Recently, several semi-supervised approaches have been proposed which work both multi-modal (Alwassel et al., 2020) and cross-model settings. (Xu et al., 2022) considers two video models with different architectures to generate pseudo-labels that are used to train each other in a cross-teaching fashion. Although semi-supervised image classification has made great strides, SSL for prompt learning is still a new and understudied issue.

## 3 Methodology

Using a pretrained vision-language model *e.g.*, CLIP (Radford et al., 2021), the aim of our proposed approach is to learn prompts in a semi-supervised setting for efficient and generalizable adaption of the model to various downstream tasks.

### 3.1 Background

**Revisiting Vision-Language Models.** We build our approach on top of a pre-trained VLM, CLIP (Radford et al., 2021), that combines a text encoder and an image encoder. Specifically, we adopt a vision transformer (ViT) (Dosovitskiy et al., 2020) based CLIP model, which is consistent with current prompt learning techniques (Zhou et al., 2022b;a). As explained below, CLIP encodes an image alongside an associated text description. The image encoder takes an image $\mathbf{I}$, splits it into $M$ fixed-size patches and embeds them into patch embeddings $\mathbf{e}_p^i$, where $p = 1, \cdots, M$ denotes spatial locations. We denote the collection of embeddings $\mathbf{E}_i = \{\mathbf{e}_p^i | p = \{1, \cdots, M\}\}$ as input to the $(i+1)^{th}$ layer $L_{i+1}$ of the vision encoder. Together with an extra

learnable classification token (`[CLS]`), the vision encoder can be compactly written as,

$$[\mathbf{x_i}, \mathbf{E_i}] = L_i([\mathbf{x_{i-1}}, \mathbf{E_{i-1}}]) \; \forall i = 1, 2, 3, ..., l \; (l \text{ is \# of layers}) \tag{1}$$

where $\mathbf{x}_i \in \mathbb{R}^d$ denote `[CLS]` embedding at $L_{i+1}$'s input space. Similarly, words from the text descriptions are sent to the text encoder to produce text embedding $\mathbf{w} \in \mathbb{R}^d$. CLIP uses a contrastive loss during training to find a combined embedding space for the two modalities. For a mini-batch of image-text pairs, CLIP maximizes the cosine similarity for each image with the matched text while minimizing the cosine similarities with all other unmatched texts.

Once the two encoders are trained, recognition can be performed by finding the similarity between an image and its textual description in the joint embedding space. In place of only the classnames a more informative natural language class description or *prompts* for these classnames can be used. Some of such carefully designed prompts found to be useful in the literature are: 'a photo of a {class}', 'a photo of a person doing {activity class}' *etc.* Given $C$ class names, the text encoder generates $C$ text embeddings $\{\mathbf{w}_c\}_{c=1}^{C}$. For a test image $\mathbf{I}$ with embedding $\mathbf{x}_l$, the prediction probability $p(y|\mathbf{I})$ is calculated as:

$$p(y|\mathbf{I}) = \frac{\exp(sim(\mathbf{x}_l, \mathbf{w}_y)/\tau)}{\sum_{c=1}^{C} \exp(sim(\mathbf{x}_l, \mathbf{w}_c)/\tau)} \tag{2}$$

where, $\tau$ is a temperature hyperparameter and $sim(.)$ denotes cosine similarity function.

**Text and Visual Prompt Learning.** To overcome the shortcomings of hand-engineered prompts, prompt learning aims to learn continuous vectors at each input token using a small amount of labeled data. Given a pre-trained model, a set of $N$ learnable vectors are introduced in the input space. In order to learn the language prompts, set of prompt vectors $\mathbf{T} = \{\mathbf{t}^i\}_{i=1}^{N}$ are introduced in the text branch of the VLM. Now, the input embeddings take the form $\{\mathbf{t^1}, \mathbf{t^2}, ..., \mathbf{t^N}, \mathbf{c}^c\}_{c=1}^{C}$, where $\mathbf{c}^c$ stands for the word embedding of the $c^{th}$ class label. Similarly, $\mathbf{V} = \{\mathbf{v^i}\}_{i=1}^{N}$ is introduced in the vision branch together with the input image tokens to learn the visual prompts. After introducing the prompts at the input layer of the vision encoder, the formulation for the $l$ layers are modified as,

$$\begin{aligned}
[\mathbf{x_1}, \mathbf{Z_1}, \mathbf{E_1}] &= L_1([\mathbf{x_0}, \mathbf{V}, \mathbf{E_0}]) \\
[\mathbf{x_i}, \mathbf{Z_i}, \mathbf{E_i}] &= L_i([\mathbf{x_{i-1}}, \mathbf{Z_{i-1}}, \mathbf{E_{i-1}}]) \; \forall i = 2, 3, ..., l
\end{aligned} \tag{3}$$

where, $\mathbf{Z_i}$ represents the features computed by the $i^{th}$ transformer layer. During training, only these task-specific text prompt ($\mathbf{T}$) and visual prompts ($\mathbf{V}$) are updated, the VLM remains unchanged.

### 3.2 `XPL`

The proposed `XPL` framework leverages on the unlabeled data in a very low labeled data regime to learn prompts that are more generalizable and enhance downstream classification performance. Though traditionally not used in prompt learning, semi-supervised approaches like *pseudo-labeling* and *consistency regularization* have demonstrated great performance in recognition (Arazo et al., 2020; Berthelot et al., 2019a;b; Chen et al., 2020a; Miyato et al., 2018; Singh et al., 2021). We propose to leverage on the huge pool of unlabeled images to shine light into the gaps between handful of labeled examples. One idea in using unlabeled data is to generate different views of the same input by augmenting it differently and force the deep network to predict the same information from the two views.

Typically, a single model trained on a handful of labeled data is used for such semi-supervised learning. In our cross-model approach we introduce an auxiliary network in addition to the primary VLM and ask them to produce the supervision for each other that in effect, encourages to learn complementary representations for the same unlabeled data. As seen in Figure 2, given an unlabeled image $\mathbf{I}$, both the networks get two distinct views $\mathbf{I}^{wk}$ and $\mathbf{I}^{str}$ of the image using a 'weak' and a 'strong' augmentation respectively. 'Weak' augmentation is standard flip-and-shift operation while RandAugment (Cubuk et al., 2020) is used for 'strong' augmentation. In our multi-modal approach, to achieve mutual collaboration between the text and visual prompts, instead of using two distinct prompts in the text and visual branches, we derive the visual prompts $\mathbf{V}$ directly from the text prompts $\mathbf{T}$ using a coupling function $\mathcal{F}(.)$, *i.e.*, $\mathbf{v}^i = \mathcal{F}(\mathbf{t}^i)$. We implement $\mathcal{F}(.)$

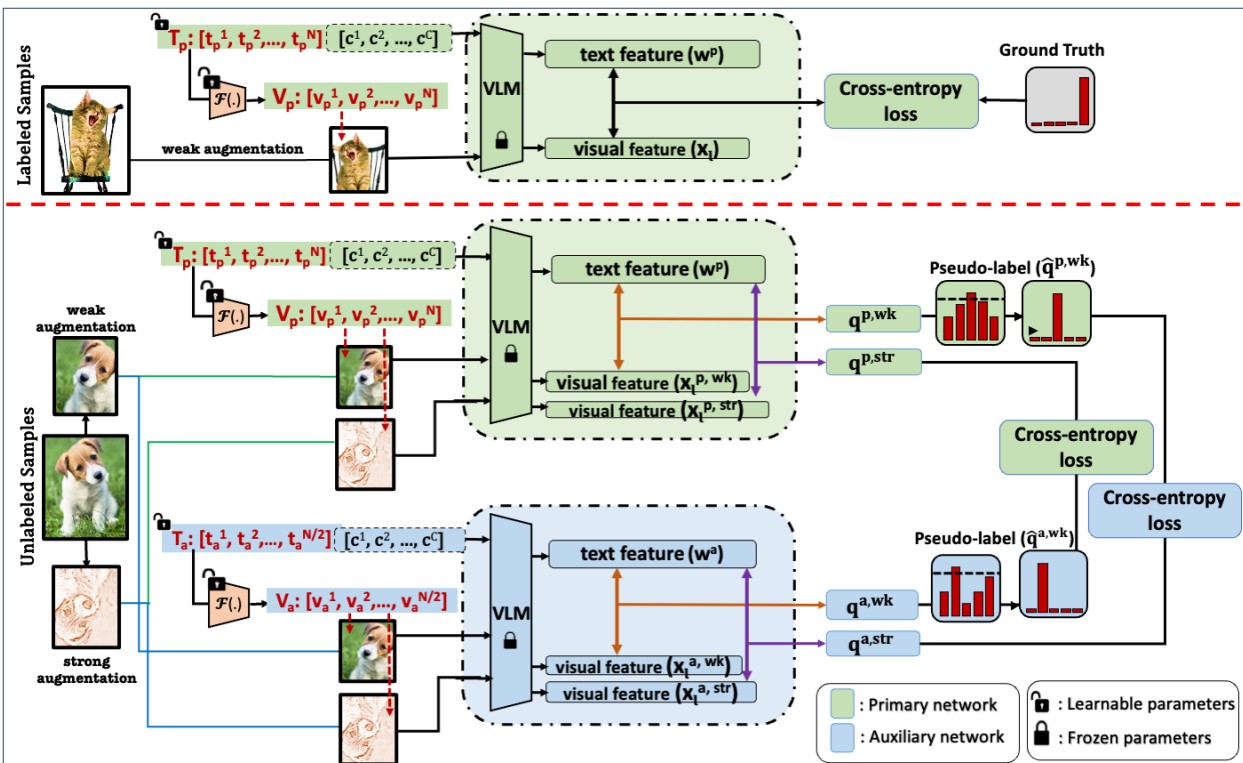

Figure 2: **Illustration of our XPL approach.** Our approach consists of primary and auxiliary networks that share the same pretrained frozen VLM. The primary network accepts text and visual prompts ($\mathbf{T}_p$ and $\mathbf{V}_p$ respectively) with $N$ tokens while the auxiliary network gets prompts ($\mathbf{T}_a$ and $\mathbf{V}_a$ respectively) with half the number of tokens. The visual prompts are generated from the textual prompts by a learnable coupling function $\mathcal{F}(.)$. At first, the prompts are learned using limited labeled data (upper portion of the red dotted line). Subsequently for the unlabeled samples (lower portion of the red dotted line), in absence of labels, prompts are trained by encouraging representations to match in both networks. This is done by minimizing the cross-entropy loss between pseudo-labels generated by the auxiliary network and the predictions made by the primary and vice versa. Given an image at test time, only the primary network is used for inference.

as a simple linear layer. For the primary network, the two prompts are denoted as $\mathbf{T}_p$ and $\mathbf{V}_p$ respectively. Similarly, the same for the auxiliary network are $\mathbf{T}_a$ and $\mathbf{V}_a$ respectively.

Given a few labeled and large amount of unlabeled data, our goal is to learn a set of prompt vectors for both $\mathbf{T}_p$ and $\mathbf{T}_a$ as well as the coupling function $\mathcal{F}(.)$. To better capitalize on the complementary information from the two networks, we propose to use prompts of different lengths (*i.e.*, different $N$) in them. Models with different number of prompt vectors exhibit markedly different behaviors in regards to category-wise performance. As the two models with different prompt lengths differ in what they learn, they can complement in generating the supervision for each other. Our primary and auxiliary networks use $N$ and $N/2$ prompt vectors respectively, *i.e.*, $\mathbf{T}_p = \{\mathbf{t}_p^i\}_{i=1}^{N}$ and $\mathbf{T}_a = \{\mathbf{t}_a^i\}_{i=1}^{N/2}$.

**Supervised Training**. A labeled image $\mathbf{I}_i$ with groundtruth class $c_i$ is only weakly augmented and passed through the model with associated text and visual prompts. Similar to Eq. 2, the prediction probabilities in the primary and auxiliary networks are given by,

$$p(y_{c_i}^p|\mathbf{I}_i) = \frac{\exp(sim(\mathbf{x}_{l,i}^p, \mathbf{w}_{c_i}^p)/\tau)}{\sum_{c=1}^{C}\exp(sim(\mathbf{x}_{l,i}^p, \mathbf{w}_c^p)/\tau)} \quad (4) \qquad p(y_{c_i}^a|\mathbf{I}_i) = \frac{\exp(sim(\mathbf{x}_{l,i}^a, \mathbf{w}_{c_i}^a)/\tau)}{\sum_{c=1}^{C}\exp(sim(\mathbf{x}_{l,i}^a, \mathbf{w}_c^a)/\tau)} \quad (5)$$

where, the superscripts $p$ and $a$ denote the primary and the auxiliary networks respectively. Given the number of labeled images $b$ in a batch, the supervised losses of the two networks are given by, $\mathcal{L}_p^{sup} = -\frac{1}{b}\sum_{i=1}^{b}\log p(y_{c_i}^p|\mathbf{I}_i)$ and $\mathcal{L}_a^{sup} = -\frac{1}{b}\sum_{i=1}^{b}\log p(y_{c_i}^a|\mathbf{I}_i)$.

**Cross-model Unsupervised Training.** For an unlabeled image $\mathbf{I}_j$, the weak and strongly augmented versions $\mathbf{I}_j^{wk}$ and $\mathbf{I}_j^{str}$ are passed through both the networks along with the learnable text and visual prompts. The final layer of the primary network's vision encoder generates two [CLS] embeddings $\mathbf{x}_{l,j}^{p,wk}$ and $\mathbf{x}_{l,j}^{p,str}$ respectively for $\mathbf{I}_j^{wk}$ and $\mathbf{I}_j^{str}$. The language encoder generates $C$ text embeddings $\{\mathbf{w}_c^p\}_{c=1}^C$. Probabilities of the weakly and strongly augmented images to belong to class $c$ are given by,

$$p(y_c^{p,wk}|\mathbf{I}_j) = \frac{\exp(sim(\mathbf{x}_{l,j}^{p,wk}, \mathbf{w}_c^p)/\tau)}{\sum_{i=1}^C \exp(sim(\mathbf{x}_{l,j}^{p,wk}, \mathbf{w}_i^p)/\tau)} \qquad (6) \qquad p(y_c^{p,str}|\mathbf{I}_j) = \frac{\exp(sim(\mathbf{x}_{l,j}^{p,str}, \mathbf{w}_c^p)/\tau)}{\sum_{i=1}^C \exp(sim(\mathbf{x}_{l,j}^{p,str}, \mathbf{w}_i^p)/\tau)} \qquad (7)$$

For all $C$ classes, these are collected as the weak and strong probability distributions, $\mathbf{q}_j^{p,wk} = [p(y_1^{p,wk}|\mathbf{I}_j), \cdots, p(y_C^{p,wk}|\mathbf{I}_j)]$ and $\mathbf{q}_j^{p,str} = [p(y_1^{p,str}|\mathbf{I}_j), \cdots, p(y_C^{p,str}|\mathbf{I}_j)]$. In a similar manner, the weak and strong probability distributions of the same image from the auxiliary network are obtained as $\mathbf{q}_j^{a,wk}$ and $\mathbf{q}_j^{a,str}$ respectively. The pseudo label from the weakly augmented image in the primary network is given by, $\hat{\mathbf{q}}_j^{p,wk}$ which is an one-hot vector with a 1 in the position of $\arg\max(\mathbf{q}_j^{p,wk})$. Likewise, $\hat{\mathbf{q}}_j^{a,wk}$ denotes the pseudo label from the weakly augmented image in the auxiliary network. The cross-model unsupervised losses are enforced as,

$$\mathcal{L}_p^u = \frac{1}{\mu b}\sum_{j=1}^{\mu b} \mathbb{1}(\max(\mathbf{q}_j^{a,wk}) \geq \rho)H(\hat{\mathbf{q}}_j^{a,wk}, \mathbf{q}_j^{p,str}), \ \mathcal{L}_a^u = \frac{1}{\mu b}\sum_{j=1}^{\mu b} \mathbb{1}(\max(\mathbf{q}_j^{p,wk}) \geq \rho)H(\hat{\mathbf{q}}_j^{p,wk}, \mathbf{q}_j^{a,str}) \qquad (8)$$

where, $\mu$ is the ratio of the number of unlabled to labeled examples in a minibatch, $\rho$ is a suitable threshold for getting the pseudolabels and $H(,)$ denotes the cross-entropy function. Overall, the loss function for learning the prompt vectors involving the limited labeled data and the unlabeled data is,

$$\mathcal{L} = \mathcal{L}_p^{sup} + \mathcal{L}_a^{sup} + \lambda(\mathcal{L}_p^u + \mathcal{L}_a^u) \qquad (9)$$

where $\lambda$ denotes a hyperparameter for scaling the relative weights of the unlabeled losses.

**Inference**. After training, we only use the primary network for inference. At test time, an image is passed through the vision encoder and the prompts along with different class names are passed through the text encoder. The class giving the maximum cosine similarity with the extracted visual features is taken as the predicted class of the test image.

## 4 Experiments

In this section, we investigate XPL and aim to address three primary research questions. *Q*1: Do prompts learned using XPL effectively leverage unlabeled data for semi-supervised classification? *Q*2: How does XPL benefit from the novel cross-model design over other methods? *Q*3: Is XPL robust towards various distribution shifts in the training data and can it generalize to unseen classes?

### 4.1 Experimental Setup

**Datasets.** We evaluate XPL on 15 diverse classification datasets, namely, (a) *Natural Images*: CropDiseases (Mohanty et al., 2016), DeepWeeds (Olsen et al., 2019), Caltech101 (Fei-Fei et al., 2004), Oxford-Pets (Parkhi et al., 2012), Flowers102 (Nilsback & Zisserman, 2008), UCF-101 (Soomro et al., 2012), ImageNet (Deng et al., 2009), StandfordCars (Krause et al., 2013); (b) *Aerial Images*: EuroSAT (Helber et al., 2019); (c) *Medical Images*: ISIC (Codella et al., 2019), ChestX (Wang et al., 2017); (d) *Illustrative Images*: Kaokore (Tian et al., 2020); (e) *Texture Images*: DTD (Cimpoi et al., 2014); (f) *Symbolic Images*: USPS (Hull, 1994); (g) *Structured Images*: Clevr-Count (Johnson et al., 2017). For experiments under domain-shift, we use the DomainNet (Peng et al., 2019) dataset.

**Baselines.** Being one of the first works in multi-modal semi-supervised prompt learning, we carefully design the baselines for a comprehensive assessment. The baselines we are going to describe are different in terms of whether or not they 1. use unlabeled data, 2. learn text or visual prompts or both and 3. use auxiliary network along with the primary network. Table 1 lists these baselines in terms of the above three aspects for a quick reference. We start with the simplest of baselines first *i.e.*, we neither use unlabeled data nor use the

| Baselines | Data | | Prompts | | Networks | |
|---|---|---|---|---|---|---|
| | Labeled | Unlabeled | Text | Visual | Primary | Auxiliary |
| TPL | ✓ | | ✓ | | ✓ | |
| VPL | ✓ | | | ✓ | ✓ | |
| MPL | ✓ | | ✓ | ✓ | ✓ | |
| TPL$^u$ | ✓ | ✓ | ✓ | | ✓ | |
| VPL$^u$ | ✓ | ✓ | | ✓ | ✓ | |
| MPL$^u$ | ✓ | ✓ | ✓ | ✓ | ✓ | |
| XTPL | ✓ | ✓ | ✓ | | ✓ | ✓ |
| XVPL | ✓ | ✓ | | ✓ | ✓ | ✓ |
| XPL (Ours) | ✓ | ✓ | ✓ | ✓ | ✓ | ✓ |

Table 1: **Baselines and our approach.** First three rows (TPL, VPL and MPL) use only the primary network and is trained using labeled data only. The baselines in the middle three rows (TPL$^u$, VPL$^u$ and MPL$^u$) are trained on both labeled and unlabeled data but still use only the primary network. TPL and TPL$^u$ learns only text prompts whereas VPL and VPL$^u$ learns only visual prompts. Baselines MPL and MPL$^u$ learns both text and visual prompts. The next two rows (XTPL and XVPL) use both labeled and unlabeled data as well as both primary and auxiliary networks, but learn either text or visual prompts. The last row (XPL) is our proposed approach which uses both labeled and unlabeled data, learns both text and visual prompts as well as uses both primary and auxiliary networks.

auxiliary network. Specifically, we aim to learn prompts by passing only labeled data through the primary network. The first such baseline is termed as *Text Prompt Learning* (TPL) which learns only textual prompts following CoOp (Zhou et al., 2022b), while the second one is *Visual Prompt Learning* (VPL) which learns only visual prompts. After these two uni-modal baselines we go to the next baseline *Multi-modal Prompt Learning* (MPL) which jointly learns both textual and visual prompts. Note that TPL, VPL, and MPL (first three rows in Table 1) operate on labeled data only and at the same time, uses only the primary network. We now leverage unlabeled data in baselines TPL$^u$, VPL$^u$, and MPL$^u$ (middle three rows in Table 1) which uses unlabeled data along with the labeled data and employ two different augmentations to the unlabeled data. However, in absence of the auxiliary network the pseudo-label for the strongly augmented image comes from its weakly augmented view in same network *i.e.*, the primary network. The next two baselines XTPL and XVPL (two penultimate rows in Table 1) use the cross-model architecture *i.e.*, both primary and auxiliary networks but learn only text prompts or only visual prompts respectively. Note that XTPL and XVPL uses both labeled and unlabeled data in doing so. Finally, the proposed approach XPL explores the use of all three aspects *i.e.*, unlabeled data, prompts from both modalities and the cross-model architecture. We show selected baselines in the main paper while compare with the rest in the appendix.

**Implementation Details.** We randomly sample 1%, 5%, and 10% of labeled data from each class and consider the rest as unlabeled, following (Sohn et al., 2020). For few-shot evaluation, we follow CoOp (Zhou et al., 2022b) to obtain the splits. For the primary network, the number of learnable tokens for the text and visual prompts is set to 16, while in the auxiliary network, it set to 8. We set the hyperparameters $\lambda = 1$, $\mu = 7$, and $\rho = 0.7$. We choose the hyperparameter values after a small-scale hyperparameter sweep which are detailed in Section 4.3. We train using a batch size of either 32 or 64 depending on the backbone. We run all experiments for 250 epochs over three random seeds and report the mean values. We use 4 NVIDIA Tesla V100 GPUs to conduct all our experiments.

## 4.2 Main Results and Comparisons

Figure 3 and 4 show the performance comparison of XPL with the baselines using ViT-B/16 backbone. In the subsequent paragraphs, we present a summary of the experimental results and key findings that motivated the development of the proposed framework.

**Multimodal Prompt Learning.** First, we discuss the superiority of multi-modal prompt learning in extracting rich information from both text and images, highlighting its advantages over unimodal approaches. As can be seen in Figure 3a, MPL outperforms TPL and VPL consistently on average *e.g.*, with 10% of the data labeled the improvements are 3.1% and 6.1%. In the extreme case of a single label per class (1-shot), MPL outperforms TPL and VPL by 1.3% and 8.2%, respectively (ref. Appendix A.1). This finding corroborates that adaptation of both the text and image encoder is more effective than adapting a single encoder.

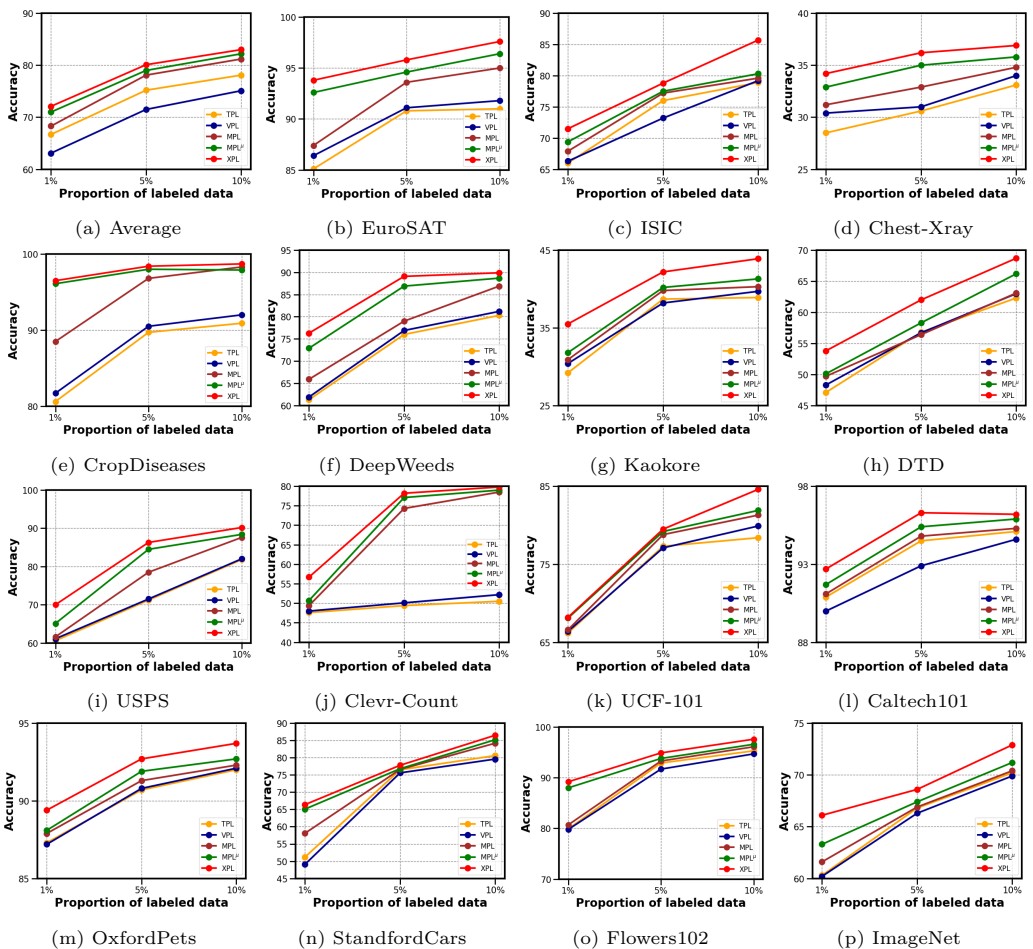

(a) Average     (b) EuroSAT     (c) ISIC     (d) Chest-Xray

(e) CropDiseases     (f) DeepWeeds     (g) Kaokore     (h) DTD

(i) USPS     (j) Clevr-Count     (k) UCF-101     (l) Caltech101

(m) OxfordPets     (n) StandfordCars     (o) Flowers102     (p) ImageNet

Figure 3: **Performance of XPL on 15 datasets with ViT-B/16** using only a small percentage of labeled training data. **XPL** leverages on the unlabeled data the most and boosts the performance across all scenarios.

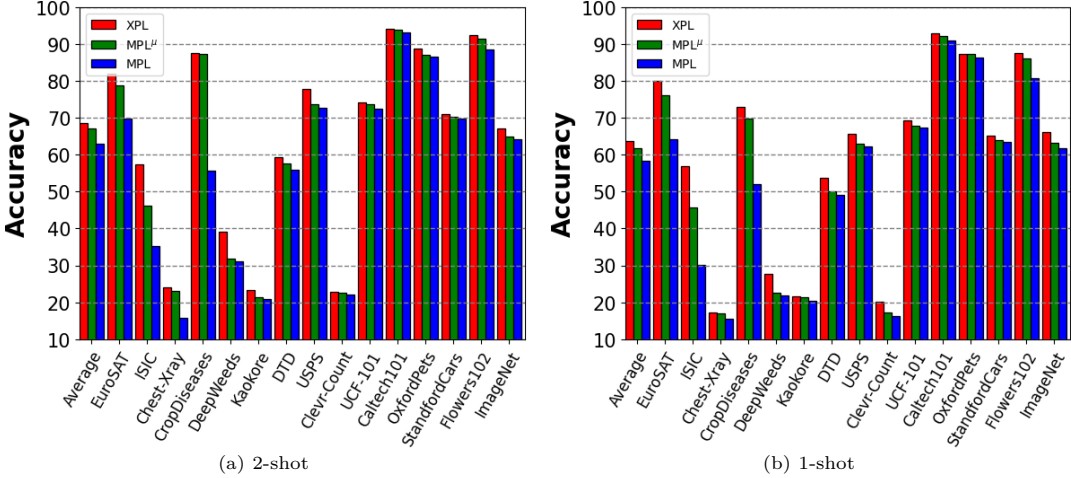

(a) 2-shot        (b) 1-shot

Figure 4: **Few-shot performance of XPL on 15 datasets with ViT-B/16.** **XPL** leverages on the unlabeled data the most and boosts the performance across all scenarios.

| | S | U | H |
|---|---|---|---|
| TPL | 74.43 | 40.53 | 49.17 |
| VPL | 74.75 | 41.43 | 50.73 |
| MPL | 77.28 | 44.67 | 54.47 |
| XPL | **80.79** | **53.32** | **62.06** |

(a) Average

| | S | U | H |
|---|---|---|---|
| TPL | 92.19 | 54.74 | 68.69 |
| VPL | 93.30 | 54.83 | 69.05 |
| MPL | 93.49 | 55.12 | 69.45 |
| XPL | **97.80** | **58.90** | **73.52** |

(b) EuroSAT

| | S | U | H |
|---|---|---|---|
| TPL | 74.30 | 19.20 | 30.51 |
| VPL | 74.32 | 20.89 | 32.61 |
| MPL | 74.90 | 26.91 | 39.59 |
| XPL | **78.40** | **80.80** | **79.58** |

(c) ISIC

| | S | U | H |
|---|---|---|---|
| TPL | 19.00 | 24.41 | 21.37 |
| VPL | 22.40 | 24.48 | 23.39 |
| MPL | 28.78 | 25.32 | 26.94 |
| XPL | **32.70** | **33.00** | **32.85** |

(d) Chest-Xray

| | S | U | H |
|---|---|---|---|
| TPL | 89.10 | 19.00 | 31.32 |
| VPL | 87.60 | 17.43 | 29.07 |
| MPL | 90.2 | 19.92 | 32.63 |
| XPL | **99.20** | **20.23** | **33.61** |

(e) CropDiseases

| | S | U | H |
|---|---|---|---|
| TPL | 76.90 | 13.80 | 23.52 |
| VPL | 80.70 | 14.76 | 24.96 |
| MPL | 81.5 | 18.41 | 30.04 |
| XPL | **89.20** | **42.80** | **57.84** |

(f) DeepWeeds

| | S | U | H |
|---|---|---|---|
| TPL | 40.70 | 36.60 | 38.54 |
| VPL | 40.76 | 36.08 | 38.28 |
| MPL | 40.90 | 37.08 | 38.90 |
| XPL | **41.10** | **37.35** | **39.14** |

(g) Kaokore

| | S | U | H |
|---|---|---|---|
| TPL | 79.44 | 41.18 | 54.24 |
| VPL | 79.00 | 42.30 | 55.10 |
| MPL | 78.70 | 42.80 | 53.80 |
| XPL | **80.18** | **43.04** | **55.98** |

(h) DTD

| | S | U | H |
|---|---|---|---|
| TPL | 88.00 | 35.10 | 50.18 |
| VPL | 89.23 | 37.08 | 52.39 |
| MPL | 94.50 | 42.20 | 58.35 |
| XPL | **96.00** | **47.20** | **63.28** |

(i) USPS

| | S | U | H |
|---|---|---|---|
| TPL | 49.10 | 21.80 | 30.19 |
| VPL | 48.70 | 15.08 | 23.03 |
| MPL | 51.60 | 17.00 | 25.57 |
| XPL | **58.40** | **22.50** | **32.48** |

(j) Clevr-Count

| | S | U | H |
|---|---|---|---|
| TPL | 64.69 | 56.05 | 60.06 |
| VPL | 66.24 | 58.96 | 62.39 |
| MPL | 75.41 | 57.76 | 65.42 |
| XPL | **88.50** | **67.70** | **76.72** |

(k) UCF-101

| | S | U | H |
|---|---|---|---|
| TPL | 98.00 | 89.80 | 93.70 |
| VPL | 96.70 | 86.70 | 91.43 |
| MPL | 98.50 | 91.60 | 94.92 |
| XPL | **99.01** | **92.52** | **95.62** |

(l) Caltech101

| | S | U | H |
|---|---|---|---|
| TPL | 92.19 | 54.74 | 68.69 |
| VPL | 93.60 | 56.10 | 70.15 |
| MPL | 94.30 | 56.80 | 70.90 |
| XPL | **97.80** | **58.80** | **73.44** |

(m) OxfordPets

| | S | U | H |
|---|---|---|---|
| TPL | 78.12 | 60.40 | 68.13 |
| VPL | 77.4 | 57.60 | 66.05 |
| MPL | 81.20 | 60.30 | 69.21 |
| XPL | **74.59** | **71.82** | **73.18** |

(n) StandfordCars

| | S | U | H |
|---|---|---|---|
| TPL | 97.60 | 59.67 | 74.06 |
| VPL | 96.60 | 57.80 | 72.32 |
| MPL | 97.54 | 63.20 | 76.70 |
| XPL | **98.24** | **69.87** | **81.66** |

(o) Flowers102

Table 2: **Comparison of `XPL` with `TPL`, `VPL` and `MPL` in generalization from base to new classes.** For a given dataset, the text and visual prompts are learned using a subset of classes (S) and evaluated on the rest of the classes (U). The results portray the strong generalizibility of our cross-model prompt learning approach. H refers to the Harmonic mean.

**Leveraging Unlabeled Data.** Here, we demonstrate the sub-optimality of disregarding unlabeled data in `MPL`, which can lead to a loss of valuable knowledge. With unlabeled data, $\text{MPL}^u$ achieves a significant gain over `MPL` specifically in the low-labeled data regime. E.g., 3.5% average improvement in 1-shot scenario can be seen in Figure 4b. It also significantly helps in challenging datasets like EuroSAT and CropDiseases, *e.g.*, performance improves by 11.9% and 17.9% respectively in 1-shot scenario, as can be seen in the same Figure.

**Cross-Model Design.** We now showcase the effectiveness of our cross-model design, which harnesses complementary knowledge from both models. As can be seen in Figures 3 and 4, `XPL` outperforms all the baselines in all the settings showing the effectiveness of the cross-model design. *E.g.*, in Figure 4b, `XPL` provides 2.9% average improvement over the strongest baseline $\text{MPL}^u$ in 1-shot scenario. Moreover, `XPL` offers a significant jump of 5% for the fine-grained DeepWeeds dataset in the same 1-shot setup validating the importance of harnessing complementary knowledge through our unique design.

**Robustness to Domain Shift in Unlabeled Data.** Adapting models to downstream data often overfits to that specific task and fails to generalize towards domain shifts. This behavior is specifically common in low-labeled data regime. For a domain $\mathcal{D}$ with a given amount of labeled ($|\mathcal{D}_l|$) and unlabeled data ($|\mathcal{D}_u|$), we define a mixture fraction $\eta$ which signifies that $\eta$ fraction of the unlabeled data ($\eta \times |\mathcal{D}_u|$) comes from a different domain $\hat{\mathcal{D}}$ while $(1 - \eta)$ fraction of it ($(1 - \eta) \times |\mathcal{D}_u|$) comes from the same domain $\mathcal{D}$. We consider two scenarios: when all the unlabeled data belong to $\mathcal{D}$ ($\eta = 0$), and when they belong to $\hat{\mathcal{D}}$ ($\eta = 1$). Table 3 shows the classification accuracy on $\mathcal{D}$ with 10% labeled training data from the same domain. We compare with the strongest baseline $\text{MPL}^u$ on three pairs of domains from the Office-31 (Saenko et al., 2010) and three pairs of domains from the DomainNet (Peng et al., 2019) dataset. As can be observed, `XPL` consistently outperforms $\text{MPL}^u$ irrespective of the domain shift for both the datasets. *E.g.*, for $\mathcal{D}$=**qdr** and $\hat{\mathcal{D}}$=**skt**, if we compare the performance of the *no domain shift* scenario ($\eta = 0$) with that of the *maximum domain shift* ($\eta = 1$), $\text{MPL}^u$'s accuracy drops by 2% (31.9% vs 29.9%) while `XPL` shows a drop of mere 0.7% (35.2% vs 34.5%) while outperforming $\text{MPL}^u$ by 4.6%. This corroborates the robustness of `XPL` towards out-of-distribution data.

**Generalization from Seen to Unseen Classes.** In Table 2, for a given dataset, we train on a subset of classes (seen) and show generalization performance to the rest of the classes (unseen). We compare `XPL` with `TPL`, `VPL` and `MPL` for accuracy on the seen classes (S) and the unseen classes (U) and their harmonic mean (H) which highlights the generalization trade-off (Xian et al., 2017). `XPL` consistently outperforms `MPL` on unseen classes across all the datasets, *e.g.* an improvement of 3.78% (55.12% vs 58.90%) and 7.68% (25.32% vs 33.00%) on EuroSAT and Chest-Xray datasets respectively. Even on the average performance

| | Office-31 | | | | | | DomainNet | | | | | |
|---|---|---|---|---|---|---|---|---|---|---|---|---|
| Method | ($\mathcal{D}$=**A**, $\hat{\mathcal{D}}$=**W**) | | ($\mathcal{D}$=**W**, $\hat{\mathcal{D}}$=**D**) | | ($\mathcal{D}$=**D**, $\hat{\mathcal{D}}$=**A**) | | ($\mathcal{D}$=**rel**, $\hat{\mathcal{D}}$=**pnt**) | | ($\mathcal{D}$=**clp**, $\hat{\mathcal{D}}$=**inf**) | | ($\mathcal{D}$=**qdr**, $\hat{\mathcal{D}}$=**skt**) | |
| (10% labeled data) | $\eta = 0$ | $\eta = 1$ | $\eta = 0$ | $\eta = 1$ | $\eta = 0$ | $\eta = 1$ | $\eta = 0$ | $\eta = 1$ | $\eta = 0$ | $\eta = 1$ | $\eta = 0$ | $\eta = 1$ |
| MPL$^u$ | 82.8 | 81.7 | 86.4 | 85.2 | 84.2 | 81.9 | 78.0 | 77.7 | 67.4 | 67.0 | 31.9 | 29.9 |
| **XPL** (Ours) | 84.7 | 84.0 | 88.2 | 87.1 | 85.5 | 84.6 | 79.1 | 78.6 | 68.0 | 67.9 | 35.2 | 34.5 |

Table 3: **Performance under domain shift in Office-31 and DomainNet.** Numbers show the accuracy on test partition of domain $\mathcal{D}$ when the models are trained with 10% labeled data from $\mathcal{D}$ and two different proportions of unlabeled data ($\eta$) between $\mathcal{D}$ and $\hat{\mathcal{D}}$. **XPL** achieves the best performance even in this challenging scenario for both the datasets.

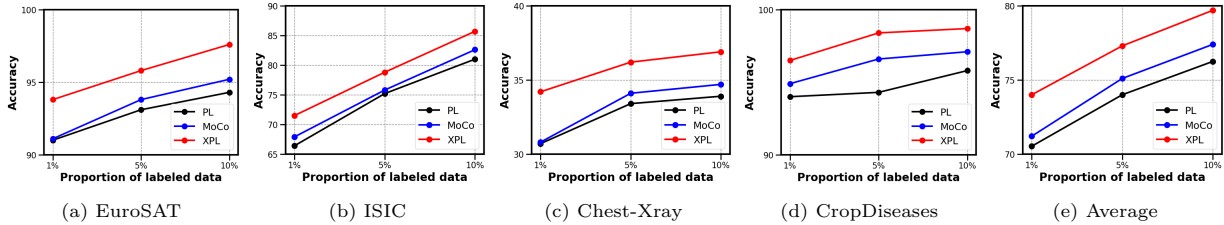

| (a) EuroSAT | (b) ISIC | (c) Chest-Xray | (d) CropDiseases | (e) Average |
|---|---|---|---|---|

Figure 6: **Comparison with self-supervised baselines.** Plots show the performance comparison of **XPL** with **PL** which uses vanilla pseudo-label training and **MoCo** which uses momentum encoder for self-supervision. **XPL** consistently outperforms both **PL** and **MoCo** across all the 4 datasets.

across all datasets, **XPL** surpasses **MPL** by a significant margin of 7.59 (62.06% vs 54.47%). Superior harmonic mean across the datasets substantiates that learning multi-modal prompts with complementary knowledge harnessed from the cross-model architecture helps improve the generalization to unseen classes.

**Different VLM Backbones.** We show the generalization of **XPL** by replacing the VLM backbones and architectures. In Figure 5, average accuracies on all datasets excluding ImageNet using CLIP ViT-B/32 (Radford et al., 2021) and DeCLIP ViT-B/32 (Li et al., 2021) backbones are reported. **XPL** consistently outperforms the baselines and obtains state-of-the-art performance for both models. *E.g.*, **XPL** outperforms **MPL**$^u$ by 1.9% (53.9% vs 52.0%) when 1% labeled data is used. This shows the effectiveness of **XPL** in harnessing complementary information even from stronger backbones like DeCLIP which has already been trained with extensive self-supervision.

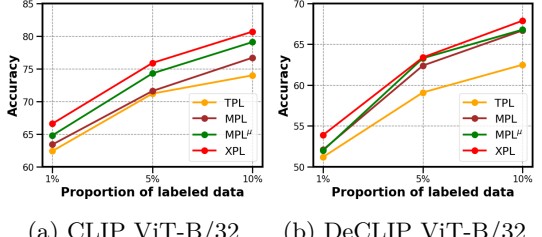

| (a) CLIP ViT-B/32 | (b) DeCLIP ViT-B/32 |
|---|---|

Figure 5: **Performance with different VLM backbones.** Plots show average accuracy using CLIP ViT-B/32 and DeCLIP ViT-B/32. **XPL** outperforms all baselines and obtains the best.

**Comparison with self-supervised baselines.** In order to assess the effectiveness of the cross-model strategy, in Figure 6 we compare **XPL** with two self-supervised baselines namely, **PL** and **MoCo**. In **PL** we have a single model and perform vanilla pseudo-label training (Lee, 2013) on the unlabeled data in addition to the supervised loss on the labeled data. Similarly, in **MoCo**, we employ the self-supervision strategy of (He et al., 2020) using momentum encoder on a single model. The performance of both **MoCo** and **PL** fails to reach that of **XPL** across the 4 datasets, *e.g.*, on average **PL** and **MoCo** shows 3.5% and 2.8% lower accuracy than **XPL** respectively for 1% labeled data (70.5% vs 71.2% vs 74.0%). This signifies the importance of the cross-model strategy in alleviating noisy and incorrect pseudo-labels by leveraging complementary information from both the networks.

### 4.3 Ablation Studies

We perform ablation studies on 4 diverse datasets (unless otherwise specified) namely EuroSAT (Helber et al., 2019), ISIC (Codella et al., 2019), ChestX (Wang et al., 2017) and CropDiseases (Mohanty et al., 2016) to test the effectiveness of different components of **XPL**. For each ablation study we report the performance on the individual dataset as well as the average performance across these 4 datasets.

**Different Prompt Lengths.** In the main experiments we learn prompts of length 16 and 8 for the primary and the auxiliary network respectively. Figure 7, shows the performance using prompts of lengths 8 and 4 respectively (XPL $(8, 4)$), on 4 datasets. As expected, using shorter prompt lengths drops the performance since the number of learnable parameters decreases. *E.g.*, on average, the accuracy drops by 3.4% (70.6% vs 74.0%) when we have 1% of labeled data. We also ran an experiment to see if using same number of prompts in two paths can harness the complimentary information as well. In two different variations of this, we used 16 prompts (XPL $(16, 16)$) and 8 prompts (XPL $(8, 8)$) in both primary and auxiliary paths. As seen, compared to the proposed approach XPL $(16, 8)$, the performance diminishes in both XPL $(16, 16)$ and XPL $(8, 8)$ showing the utility of using different prompt lengths in primary and auxiliary models. Lastly, we tried to see if increasing the ratio of the number of prompt vectors in the two paths helps more. As seen, if we use 32 and 8 prompts in the two paths (XPL $(32, 8)$) the performance diminishes which is possibly due to a large mismatch in the capacities of the two paths.

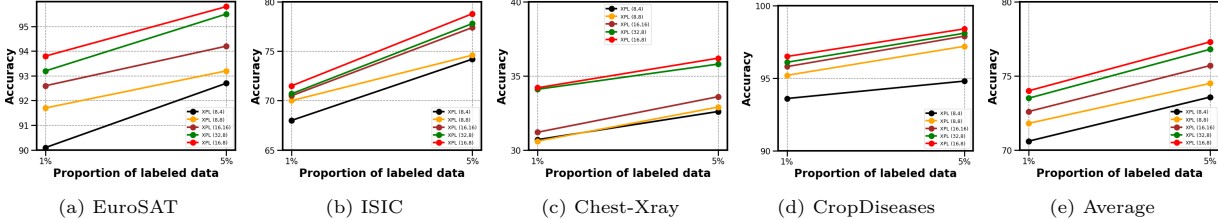

(a) EuroSAT     (b) ISIC     (c) Chest-Xray     (d) CropDiseases     (e) Average

Figure 7: **Different Prompt Lengths.** Plots show the accuracy curves for XPL using different prompt lengths. XPL $(M, N)$ learns prompts of length $M$ and $N$ for the primary and auxiliary network.

**Varying Prompt Positions.** In this ablation study, we observe the effect of changing the relative positions of the class token [CLS] and the prompt vectors, as an additional attribute instead of the length of the learnable prompts. In XPL, the [CLS] token was placed at the 'end' of the learnable prompt vectors for both primary and auxiliary branches. Here, we consider two setups with the same prompt lengths of 16 for both the branches: (1) [CLS] token is positioned at the end *i.e.*, after the prompt vectors in the primary branch, while at the beginning in the auxiliary ('beg', 'end'); (2) [CLS] token is positioned at the middle of the prompt vectors in the primary branch, while at the end in the auxiliary ('mid', 'end'). As can be observed over all the 4 datasets across 2 different proportions of labeled data (1% and 5%), changing the class token positions does not distinctively affect the performance of our approach. Rather we have seen that the lengths of the prompts play a more significant role in the cross-model approach. The use of different prompt lengths harnesses the most complementary information and provides the best performance.

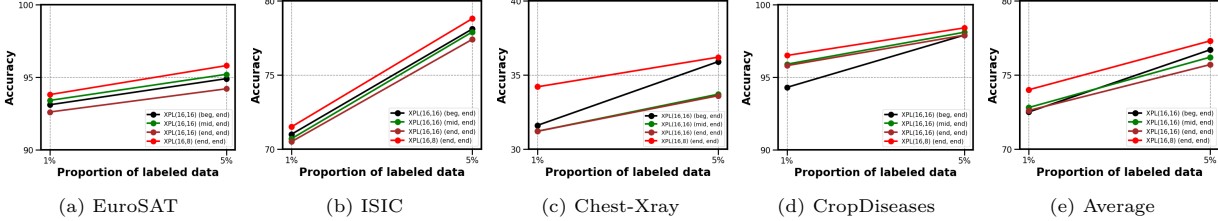

(a) EuroSAT     (b) ISIC     (c) Chest-Xray     (d) CropDiseases     (e) Average

Figure 8: **Varying the position of prompts in the two branches.** Plots show the accuracies for XPL after appending the class tokens, [CLS], at different positions of of the learnable prompt vectors for both primary and auxiliary branches. XPL $(M, N)$ $(pos1, pos2)$ learns prompts of length $M$ and $N$ with the [CLS] token appended in the $pos1$ and $pos2$ positions for the primary and auxiliary network respectively. Here *beg*, *mid* and *end* refers to putting the [CLS] token in the beginning, mid

**Effect of Hyperparameters.** In this Figure 10 shows the average performance over 4 datasets, EuroSAT, ISIC, Chest-Xray and Cropdiseases. First, we vary $\lambda$ (ref. Eq. 9) to 0.5, 1.0 and 2.0 in XPL and obtain the best performance when all the losses are equally weighed (i.e. $\lambda = 1.0$) and is used in our experiments. The ratio

(a) $\lambda$     (b) $\mu$     (c) $\rho$

Figure 10: **Effect of Hyperparameters.** Plots analyze the average performance across 4 datasets by varying hyperparameters $\lambda$, $\mu$, and $\rho$.

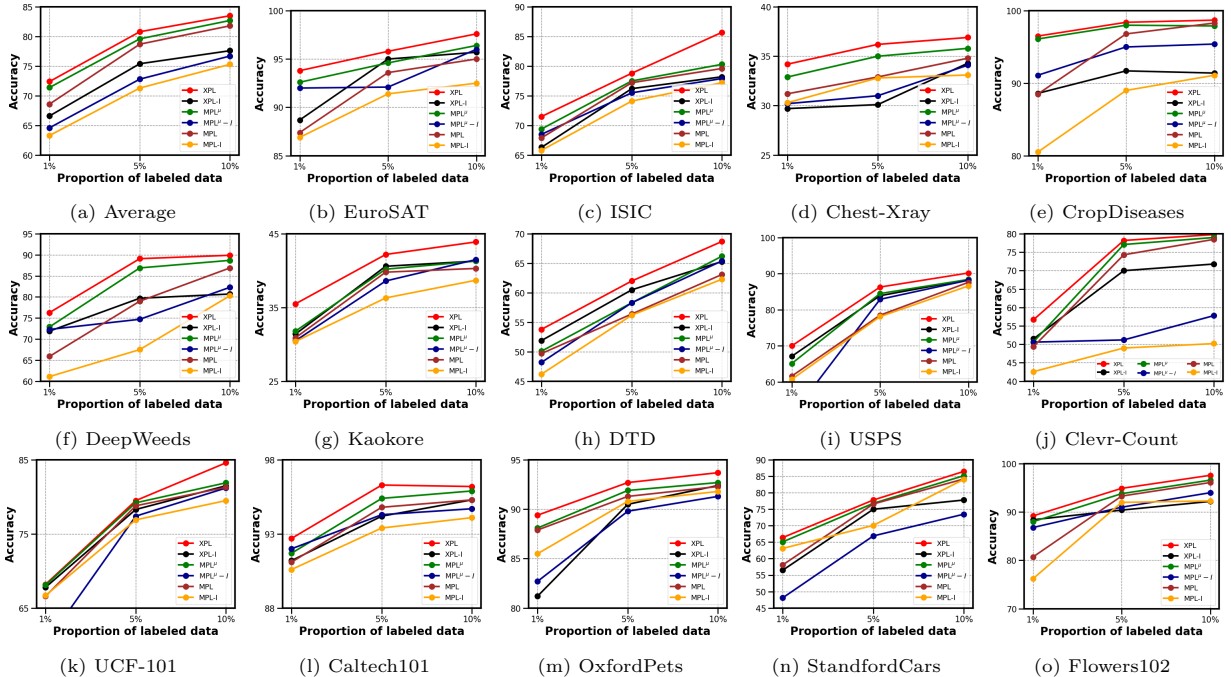

Figure 9: **Effect of coupling function** $\mathcal{F}(.)$**.** Accuracy across 14 datasets shows that mutual collaboration between the text and visual prompts through $\mathcal{F}(.)$ is necessary for improved performance.

of unlabeled to labeled data $\mu$ (ref. Eq. 8) is important in deciding the performance. We vary $\mu$ to 5, 7, and 9. The performance increases with higher values of $\mu$, however, scaling up $\mu$ often requires high computational resources. We observe negligible improvement beyond $\mu = 7$ and hence use that for XPL. We also vary the pseudo-label threshold $\rho$ (ref. Eq. 6, 7) to 0.6, 0.7, and 0.95. We obtain the best performance at $\rho = 0.7$ and use it for all the experiments.

**Effect of coupling function** $\mathcal{F}(.)$ As shown in Figure 2, we use a coupling function $\mathcal{F}(.)$ to ensure mutual collaboration between the text and visual prompts (hence the encoders). In order to study its effect, we remove $\mathcal{F}(.)$ and independently learn the text and visual prompts for XPL, MPL$^u$, MPL, resulting in methods XPL-I, MPL$^u$-I, and MPL-I respectively. We show the individual performances of these baselines in all 14 the datasets along with the average performance for 1%, 5% and 10% proportions of labeled data in Figure 9 and 2-shot and 1-shot performances in Appendix (Figure 15). As can be observed in both the settings, removing $\mathcal{F}(.)$ decreases the average performance, *e.g.*, 5.8% (72.4% vs 66.6%) for XPL with 1% labeled data and hence ensuring that mutual coherence between the text and visual prompts is crucial for better performance.

# 5 Conclusion

We present XPL, a novel cross-model framework for multi-modal, semi-supervised prompt learning towards parameter-efficient adaptation of large pretrained VLMs to different downstream tasks. We identify that directly using the same adaptation model to produce confident pseudo-labels for the unlabeled data may miss crucial category-wise information. A novel cross-model semi-supervision is pioneered to leverage the complimentary knowledge learned by models with different length prompts significantly improving the performance. We demonstrate the effectiveness of our proposed approach on fourteen benchmark datasets, outperforming several competing methods. Our research can help reduce burden of collecting large-scale supervised data in many real-world vision applications by transferring knowledge from large pretrained VLMs. Limitations of our research are difficult to predict, however, using more data, albeit unlabeled may mean more computation, but this comes with a lot of savings in human annotation efforts for a similar performance gain in a fully supervised setup.

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

# A Appendix

Here we provide additional experiments and visualizations on the datasets to further explore the XPL approach. These are summarized in the following Table 4.

| Section | Content |
|---------|---------|
| A.1 | Leveraging Unlabeled Data for Uni-modal baselines |
| A.2 | XPL in Uni-modal setting |
| A.3 | Effect of Coupling Function in Fewshot Setting $\mathcal{F}(.)$ |
| A.4 | Different VLM Backbones |
| A.5 | Qualitative Results |
| A.6 | t-SNE Visualizations |

Table 4: Overview of Appendix.

**Code.** Please refer to **XPL_code.zip** in the appendix material for our code submission. We will make the code public.

## A.1 Leveraging Unlabeled Data for Uni-modal baselines

In this section of the appendix, we demonstrate the sub-optimality of disregarding unlabeled data for the uni-modal baselines TPL and VPL in a similar manner as shown for the multi-modal baseline in section 4.2 of the main paper. Both $\text{TPL}^u$ and $\text{VPL}^u$ obtains a significant gain in performance as can be seen in Figure 11. On average, $\text{TPL}^u$ helps to perform better by 3% than TPL, whereas, $\text{VPL}^u$ shows 2% gain in accuracy over VPL, when using only 1% labeled data. Similar trend is also observed for 2-shot and 1-shot scenarios as shown in Figure 12. XPL retains the supremacy across all the baselines in both few-shot setting and well as in low percentages of labeled data.

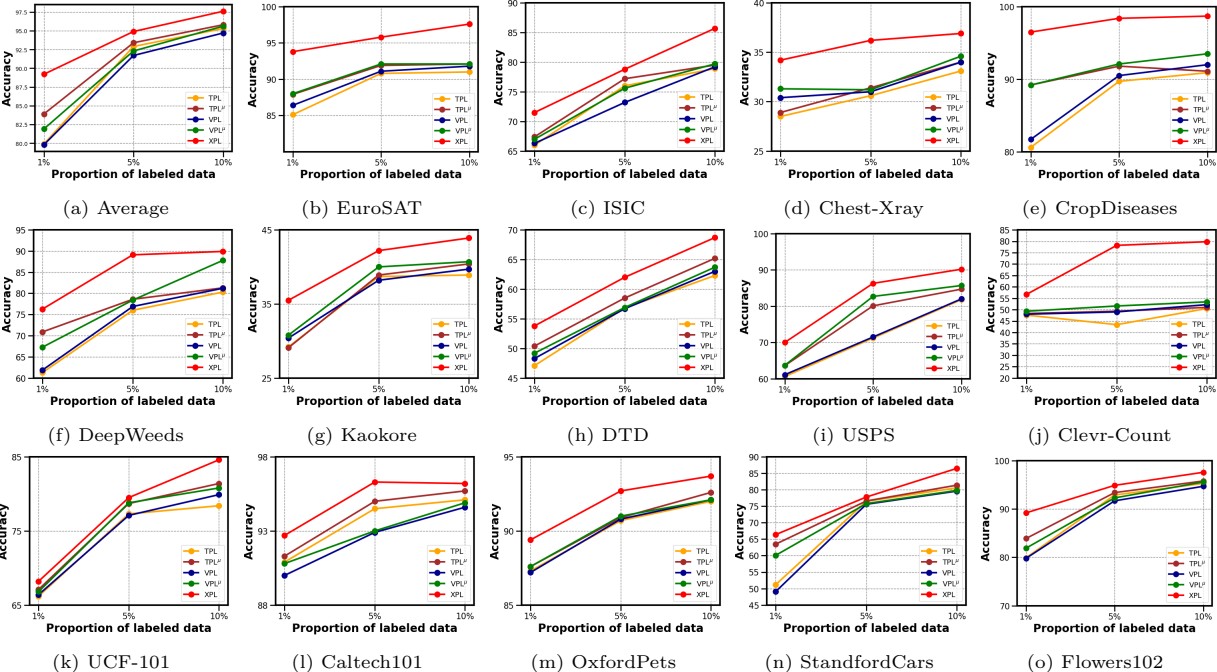

Figure 11: **Performance of TPL, $\text{TPL}^u$, VPL, $\text{VPL}^u$ and XPL on 14 datasets with ViT-B/16** using only a small percentage of labeled training data. The uni-modal baselines $\text{TPL}^u$ and $\text{VPL}^u$ leverage on the unlabeled data to obtain performance gain over TPL and VPL respectively across all scenarios. XPL leverages on the unlabeled data the most and obtains maximum boost in the performance.

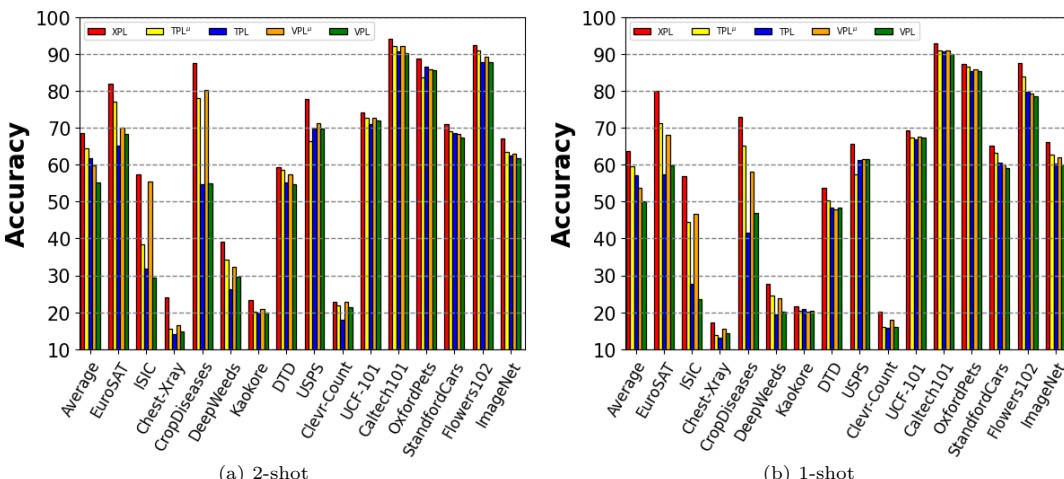

(a) 2-shot

(b) 1-shot

Figure 12: **Few-shot performance of TPL, TPL$^u$, VPL, VPL$^u$ and XPL on 15 datasets with ViT-B/16.** Even in 2-shot and 1-shot scenarios the uni-modal baselines TPL$^u$ and VPL$^u$ leverage on the unlabeled data to obtain performance gain over TPL and VPL respectively across all scenarios. XPL leverages on the unlabeled data the most in the few-shot setting to give the highest performance.

## A.2  XPL in Uni-modal Setting

Here, we showcase the importance of multi-modal prompt learning to extract richer information from both text and images compared to the unimodal approaches. As can be seen in Figure 13, we consider two baselines XTPL and XVPL, having only text prompts and only visual prompts respectively as the two uni-modal variants of XPL. In both low proportions labeled data (Figure 13) and few-shot settings (Figure 14), XPL obtains the most hike in accuracy over both XTPL and XVPL. Even for challenging datasets like DeepWeeds (Olsen et al., 2019) (refer Figure 13f) and Clevr-Count (Johnson et al., 2017) (refer Figure 13j), XPL shows the supremacy in performance by almost 10% and 35% gains respectively when using only 5% labeled data.

## A.3  Effect of coupling function $\mathcal{F}(.)$ in fewshot setting

Here we explore the effects of using the coupling function $\mathcal{F}(.)$ on the individual performances of all 14 the datasets in the fewshot scenarios. We compare the performances of XPL, MPL$^u$, MPL with XPL-I, MPL$^u$-I, and MPL-I in 2-shot and 1-shot settings respectively in Figure 15. As observed, following similar trend in Figure 9 of the main paper, using $\mathcal{F}(.)$ boosts the average performance across all the datasets.

## A.4  Different VLM Backbones

We have shown the supremacy of XPL with other VLM architectures, CLIP ViT-B/32 (Radford et al., 2021) and DeCLIP ViT-B/32 (Li et al., 2021) in Figure 5 of the main paper. Here, we illustrate those plots providing the variation in performance across the individual 14 datasets with low proportions of training data for CLIP ViT-B/32 in Figure 16 and DeCLIP ViT-B/32 in Figure 18. The average plots from the main paper (refer Figure 5) have also been included in Figures 16a and 18a for CLIP ViT-B/32 and DeCLIP ViT-B/32 respectively, for reference. We explore the performances with the two VLM backbones under few-show setting as well and plot the accuracies in Figures 17 for CLIP ViT-B/32 and 19 DeCLIP ViT-B/32 respectively.

## A.5  Qualitative Results

Figure 20 shows the qualitative examples for comparing the performance of XPL with the baselines of TPL, VPL, MPL and also the next-best MPL$^u$. As can be seen, XPL proves its supremacy in identifying diverse image samples such as different landscapes in EuroSAT (Helber et al., 2019) (Figure 20a), flower types in Flowers102 (Nilsback & Zisserman, 2008) (Figure 20b) and also animals in OxfordPets (Parkhi et al., 2012) (Figure 20c).

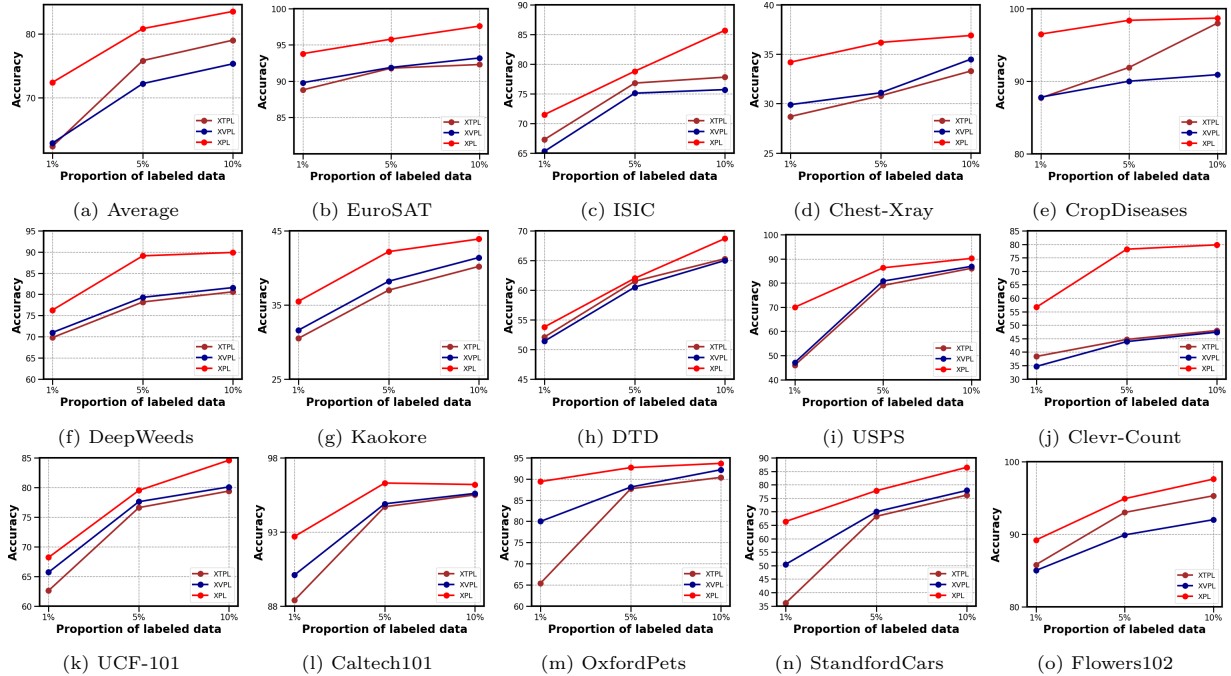

Figure 13: **Performance of XTPL, XVPL and XPL on 14 datasets with ViT-B/16** using only a small percentage of labeled training data. **XPL** obtains higher performance gain over **XTPL** and **XVPL** respectively across all scenarios. The adaptation of both the text and image encoder in **XPL** is more effective than adapting a single encoder as in **XTPL** and **XVPL**.

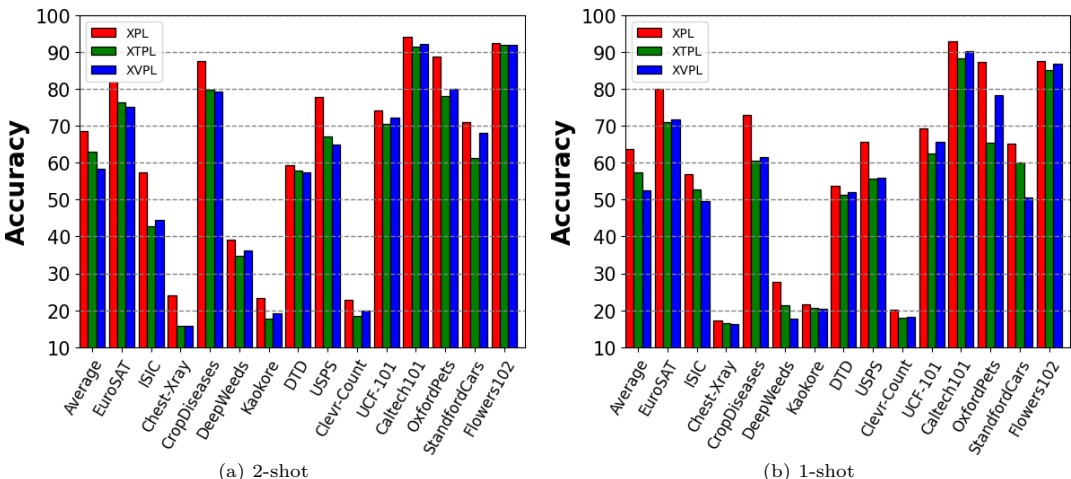

Figure 14: **Few-shot performance of XTPL, XVPL and XPL on 14 datasets with ViT-B/16. XPL** obtains higher performance gain over **XTPL** and **XVPL** respectively in both 2-shot and 1-shot setting across all datasets. The adaptation of both the text and image encoder in **XPL** is more effective than adapting a single encoder as in **XTPL** and **XVPL**.

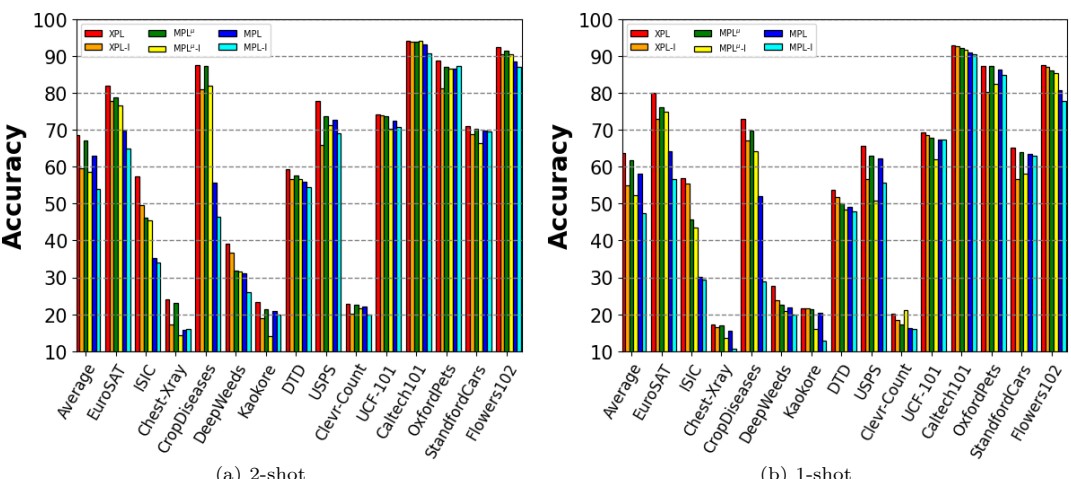

(a) 2-shot

(b) 1-shot

Figure 15: **Effect of coupling function** $\mathcal{F}(.)$**.** Fewshot performance across 14 datasets also shows mutual collaboration through $\mathcal{F}(.)$ is necessary for performance gain.

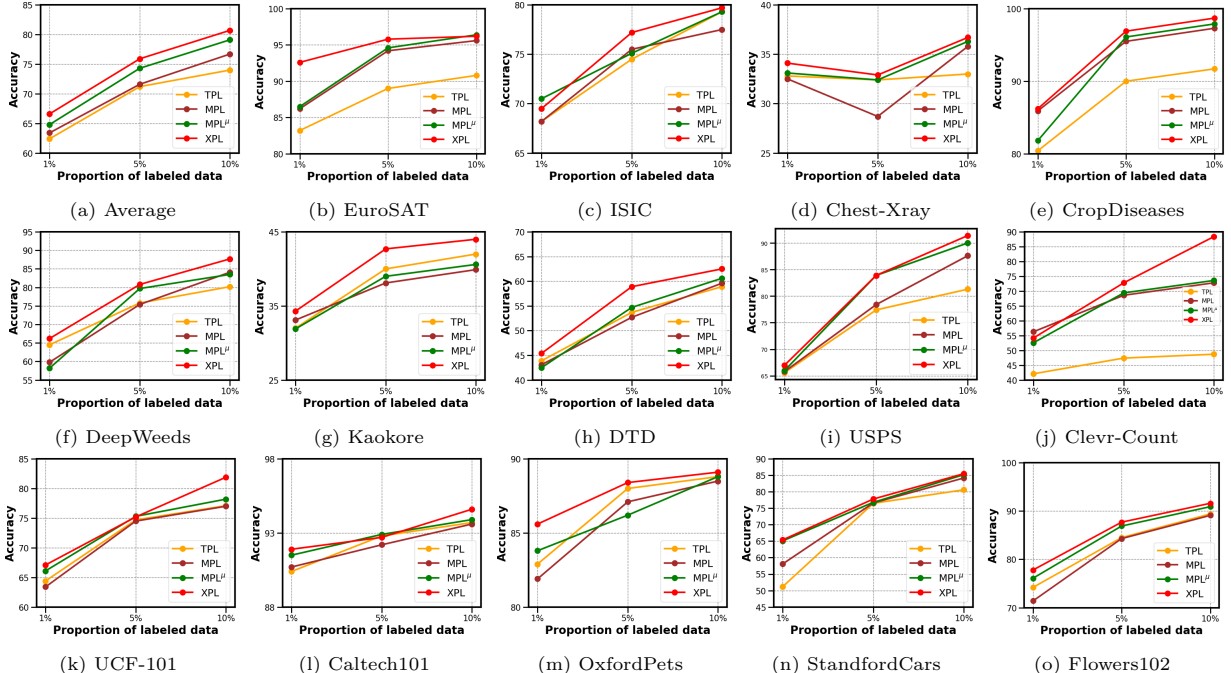

(a) Average (b) EuroSAT (c) ISIC (d) Chest-Xray (e) CropDiseases

(f) DeepWeeds (g) Kaokore (h) DTD (i) USPS (j) Clevr-Count

(k) UCF-101 (l) Caltech101 (m) OxfordPets (n) StandfordCars (o) Flowers102

Figure 16: **Performance using CLIP ViT-B/32.** Plots show accuracy across 14 datasets using CLIP ViT-B/32. **XPL** outperforms all the baselines for each dataset and obtains the best performance.

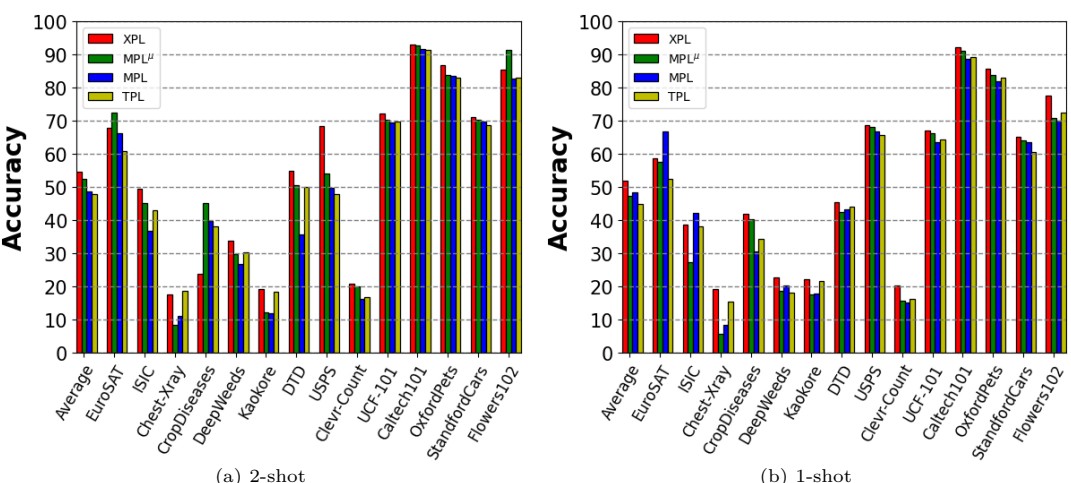

Figure 17: **Few-shot performance using CLIP ViT-B/32.** Plots show accuracy across 14 datasets. `XPL` outperforms all the baselines for each dataset and obtains the best performance.

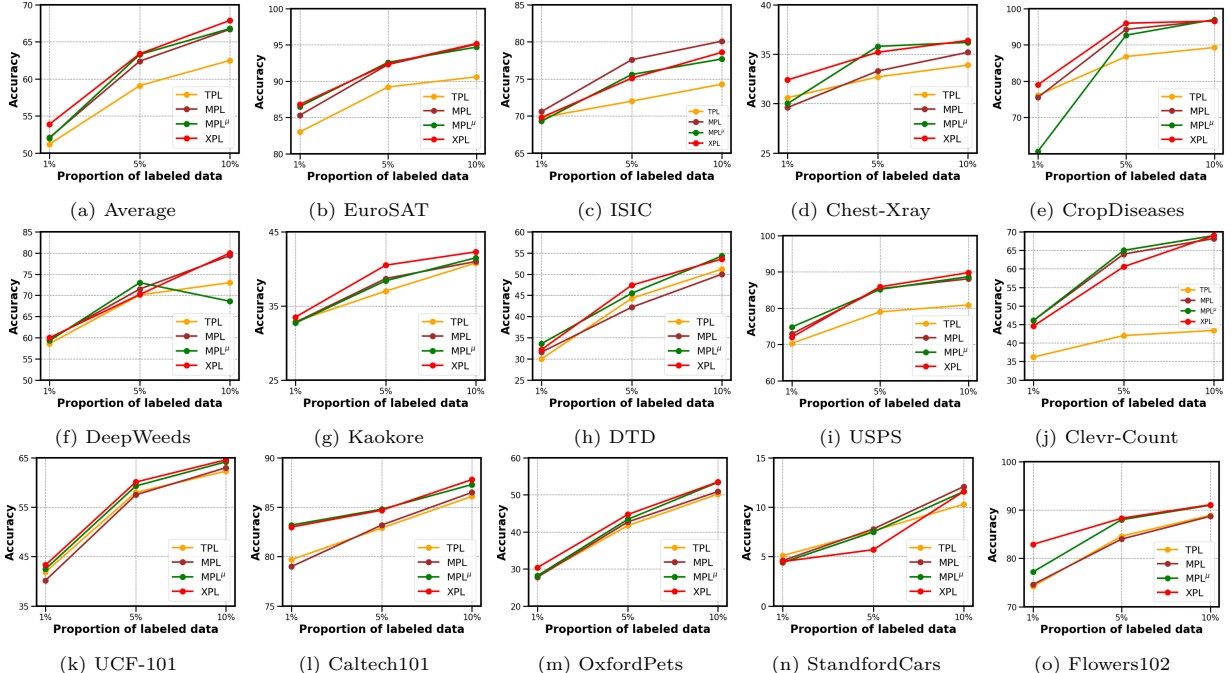

Figure 18: **Performance using DeCLIP ViT-B/32.** Plots show accuracy across 14 datasets using CLIP ViT-B/32. `XPL` outperforms all the baselines for each dataset and obtains the best performance.

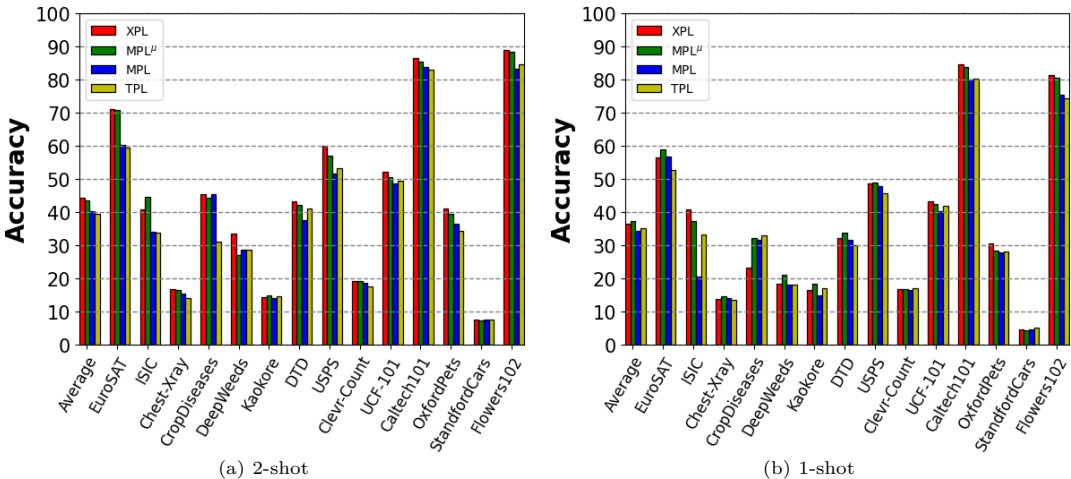

Figure 19: **Few-shot performance using DeCLIP ViT-B/32.** Plots show accuracy across 14 datasets. `XPL` outperforms all the baselines for each dataset and obtains the best performance.

## A.6 t-SNE Visualizations

Figure 21 shows the t-SNE visualizations of `XPL` along with the next-best baseline `MPL`$^u$ and also uni-modal `VPL`, `TPL` across 3 datasets of EuroSAT (Helber et al., 2019) (Figure 21a), Flowers**102** (Nilsback & Zisserman, 2008) (Figure 21b) and OxfordPets (Parkhi et al., 2012) (Figure 21c). Inspite of diverse datasets, `XPL` portrays the most consistent clustering and class-wise discriminative acoss all the 3 datasets, showing the efficacy of our cross-model approach in learning discriminative features in a multi-modal setting.

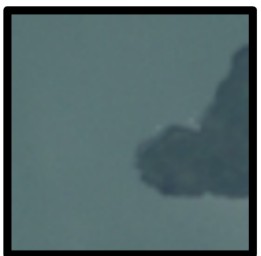 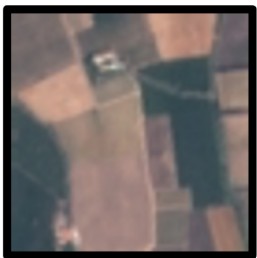 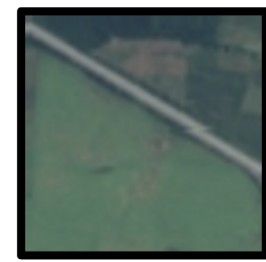 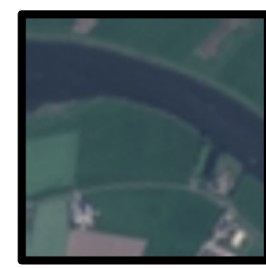

**TPL: AnnualCrop**
**VPL: Highway**
**MPL: Pasture**
**MPLᵘ: River**
**XPL: SeaLake**

**TPL: Highway**
**VPL: Industrial**
**MPL: River**
**MPLᵘ: Pasture**
**XPL: PermanentCrop**

**TPL: Residential**
**VPL: Industrial**
**MPL: Pasture**
**MPLᵘ: Highway**
**XPL: Highway**

**TPL: PermanentCrop**
**VPL: Pasture**
**MPL: HerbaceousVegetation**
**MPLᵘ: Forest**
**XPL: River**

(a) EuroSAT

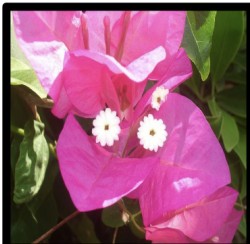 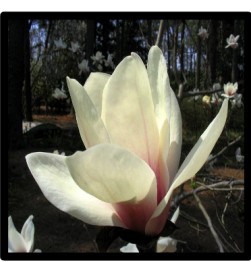 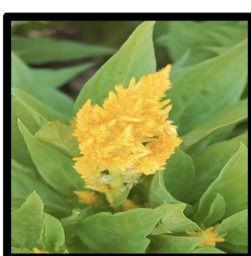 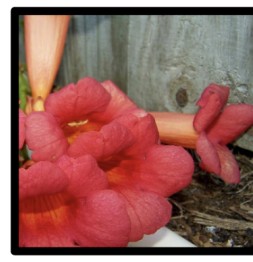

**TPL: cyclamen**
**VPL: siam tulip**
**MPL: toad lily**
**MPLᵘ: bougainvillea**
**XPL: bougainvillea**

**TPL: tree mallow**
**VPL: passion flower**
**MPL: magnolia**
**MPLᵘ: magnolia**
**XPL: magnolia**

**TPL: balloon flower**
**VPL: sweet william**
**MPL: corn poppy**
**MPLᵘ: daffodil**
**XPL: prince of wales feathers**

**TPL: blackberry lily**
**VPL: balloon flower**
**MPL: wallflower**
**MPLᵘ: foxglove**
**XPL: trumpet creeper**

(b) Flowers102

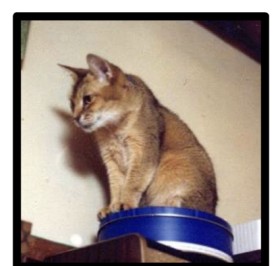 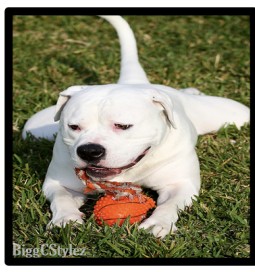 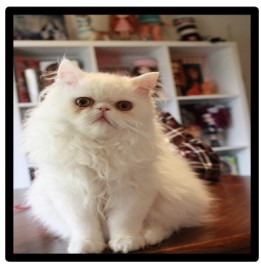 

**TPL: beagle**
**VPL: miniature_pinscher**
**MPL: chihuahua**
**MPLᵘ: Bombay**
**XPL: Abyssinian**

**TPL: Maine_Coon**
**VPL: shiba_inu**
**MPL: pug**
**MPLᵘ: boxer**
**XPL: american_bulldog**

**TPL: Bengal**
**VPL: havanese**
**MPL: pomeranian**
**MPLᵘ: Persian**
**XPL: Persian**

**TPL: chihuahua**
**VPL: Ragdoll**
**MPL: boxer**
**MPLᵘ: Pug**
**XPL: Pug**

(c) OxfordPets

Figure 20: **Qualitative examples comparing XPL with TPL, VPL and MPL baselines**. We compare the performances on 3 datasets, EuroSAT (Helber et al., 2019), Flowers102 (Nilsback & Zisserman, 2008) and OxfordPets (Parkhi et al., 2012) trained using 1% labeled data with CLIP ViT-B/16. The correct predictions are marked in green while the incorrect predictions have been marked red.

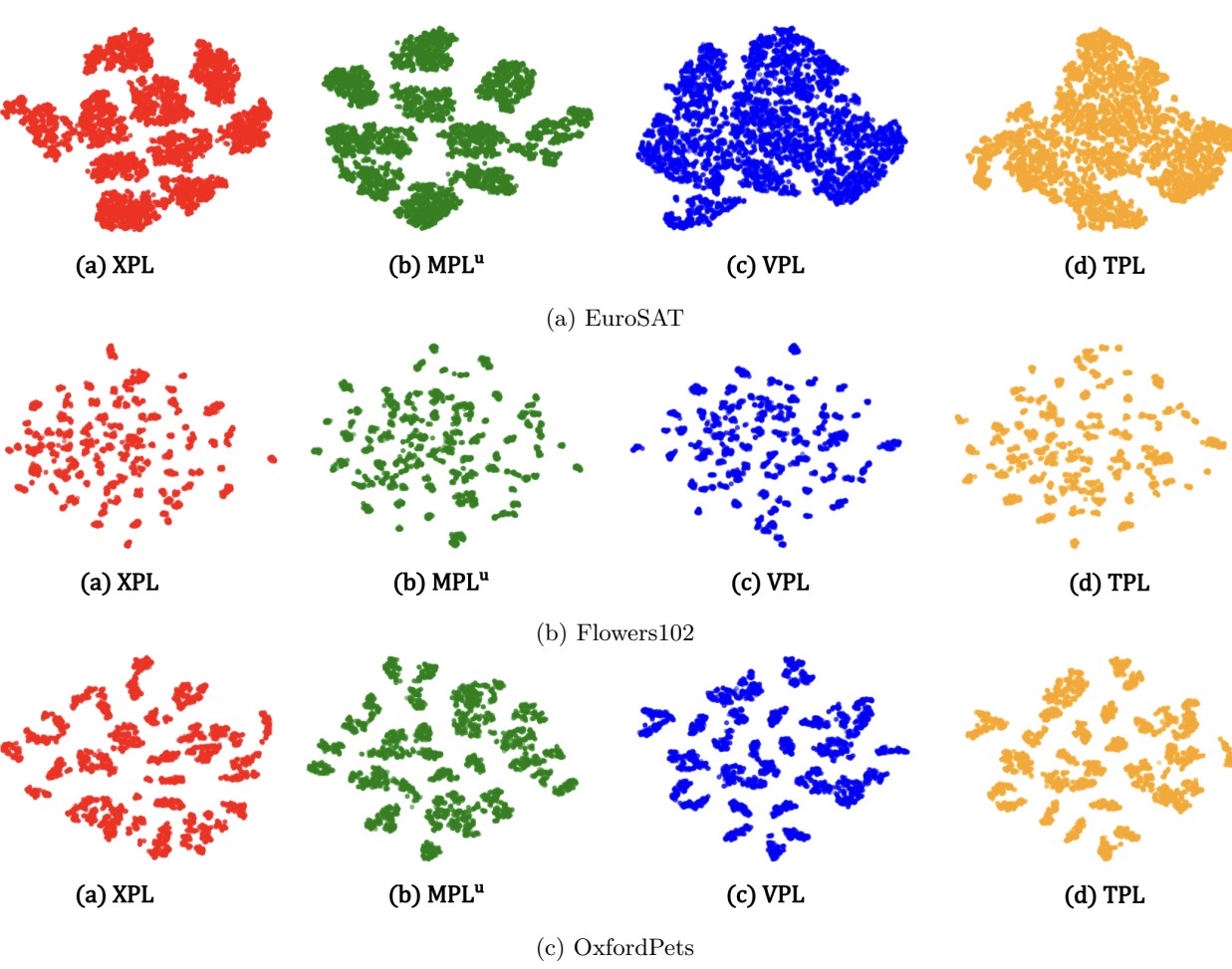

Figure 21: **Feature Visualization using t-SNE.** Figure shows the t-SNE visualizations for XPL along with 3 different baselines of MPL$^u$, VPL and TPL on 3 diverse datasets, EuroSAT (Helber et al., 2019), Flowers102 (Nilsback & Zisserman, 2008) and OxfordPets (Parkhi et al., 2012) trained using 1% labeled data with CLIP ViT-B/16. XPL forms most consistent clustering and performs better at classwise discrimination across the 3 diverse datasets. the Best viewed in color.

