# OpenReview forum: "XPL: A Cross-Model framework for Semi-Supervised Prompt Learning in Vision-Language Models"
_TMLR — Accepted by TMLR_

### Review · Reviewer_G6Rs · 2024-03-17

**Summary Of Contributions:**

- Introduce XPL, a cross-model framework for semi-supervised prompt learning in VLMs.
- Propose a semi-supervised prompt learning approach, making prompts invariant to diverse views of unlabeled samples.
- Utilize various image augmentations and prompt length variations to generate multiple views of unlabeled samples.

**Audience:**

Yes

**Claims And Evidence:**

Yes

**Requested Changes:**

Can author visualize the learned text and visual prompts? Visualizations can enhance the interpretability of the model's behavior, aiding in understanding its inner workings and potential biases. Providing clear and insightful visualizations could be crucial for readers to assess the validity and effectiveness of the proposed approach.

**Strengths And Weaknesses:**

- Innovative Approach: The framework introduces a novel method for leveraging unlabeled data in vision-language models, potentially improving model performance without additional labeled data.
- Comprehensive Analysis and Study: The framework includes a thorough examination of various aspects such as the impact of different augmentation techniques, prompt lengths, and semi-supervised learning strategies. This comprehensive analysis provides valuable insights into the effectiveness of the proposed approach and contributes to a deeper understanding of the underlying mechanisms.

---

> ### Author Response · Authors · 2024-04-06
> **Response to Reviewer G6Rs**
>
> We thank Reviewer G6Rs for finding our work innovative and commending the comprehensive analysis over 16 datasets.
>
> We also thank the Reviewer of showing interest in the aspect of visualization. Indeed, visualization can provide insights on the interpretability of the model's behavior. This query of the reviewer is addressed in Sections A.6 and A.5 of the appendix.
> In A.6 we show the t-SNE visualizations of XPL along with the next-best baseline MPL$^u$ and also uni-modal VPL, TPL across 3 datasets of EuroSAT, Flowers102 and OxfordPets. These visualizations interpret how the complimentary training of the model using different prompts in both textual and visual contexts exhibit the most consistent clustering and class-wise discriminative acoss all the 3 diverse datasets. XPL using cross-model training with multi-modal prompts, learns better discriminative features to boost the model performance. Also in A.5, we show the qualitative examples for comparing the performance of XPL with the baselines of TPL, VPL, MPL and also the next-best MPL$^u$. Here also, XPL proves its supremacy in identifying diverse visual samples, such as different landscapes in EuroSAT, flower types in Flowers102 and also animals in OxfordPets.

---

### Review · Reviewer_W2o1 · 2024-03-21

**Summary Of Contributions:**

The paper introduces XPL, a semi-supervised framework for learning prompts in vision-language models (VLMs), leveraging both labeled and unlabeled data. It identifies the limitations of prior supervised prompt learning approaches, which depend heavily on labeled data, and proposes utilizing the vast amounts of available unlabeled data to improve learning efficiency and model performance. XPL employs a novel cross-model design where two models with different prompt lengths learn complementary knowledge from unlabeled data, using a coupling function to ensure consistency between text and visual prompts. This approach not only improves the adaptability of VLMs to various downstream tasks but also enhances their generalization ability, as demonstrated by extensive experiments across multiple datasets. The results show that XPL significantly outperforms existing methods, especially in low-data regimes, showcasing its effectiveness in utilizing unlabeled data and its potential to reduce the reliance on large labeled datasets for training VLMs.

**Audience:**

Yes

**Broader Impact Concerns:**

This paper will contribute a broad impact to the vision-language community for this semi-supervised learning metho.

**Claims And Evidence:**

Yes

**Requested Changes:**

I hope the authors could address my concerns above.

**Strengths And Weaknesses:**

# Strengths:
1. Interesting General Idea:
The concept of utilizing semi-supervised training to enhance the learning of prompts is intriguing and represents a novel approach in the field. This methodology could pave the way for more efficient learning paradigms, especially in scenarios where labeled data is scarce but unlabeled data is abundant. The idea is not only innovative but also very timely, given the increasing interest in making machine learning models more data-efficient and capable of learning from limited supervision.

2. Comprehensive Experiments and Ablation Studies:
The thoroughness of the experiments and the inclusion of ablation studies significantly strengthen the paper. These studies provide valuable insights into the impact of various components of the proposed method, offering a clear understanding of what contributes most to its effectiveness. Such comprehensive experimentation is crucial for validating the robustness and generalizability of the proposed approach. It also helps in establishing a benchmark for future work, encouraging more transparent and replicable research in the community.


# Weaknesses:

1. Inaccuracy in Abstract regarding Prompt Learning:
The statement in the abstract that prompt learning focuses solely on learning continuous soft prompts overlooks significant existing work on learning discrete prompts. This oversight might mislead readers about the scope and diversity of prompt learning methodologies. Acknowledging the existence and relevance of discrete prompt learning methods [1, 2], although in the NLP domain, is important for providing a holistic view of the field and ensuring that the contributions of the paper are accurately positioned within the broader research landscape.

2. Lack of Distinction between Equations 4 and 5:
The similarity between Equations 4 and 5, with the only difference being the subscript of primary/auxiliary networks, raises questions about the novelty or necessity of presenting both as separate equations. This could potentially confuse readers or dilute the perceived contribution of the mathematical formulation. Clarifying the specific roles or implications of these equations in the context of the proposed method could help in underscoring their relevance.

3. Questioning the Advantage of the “Networks” Column in Table 1:
The presentation of the “Networks” column as an advantage in Table 1 might be misleading, as employing an auxiliary network could introduce additional computational overhead and complexity, which might not be preferable in all contexts. Efficiency is a critical consideration in the deployment of machine learning models, especially in resource-constrained environments. Highlighting this aspect could offer a more balanced view of the trade-offs involved in the proposed method.

4. Exclusion of Baseline Comparisons in Sections 4.2 and 4.3:
The absence of a baseline comparison that lacks any visual/text prompt in Sections 4.2 and 4.3 is a missed opportunity to demonstrate the significance and added value of the prompt learning methods being proposed. Including such a baseline would provide a clearer picture of the performance improvements attributable to the introduction of prompts, thereby strengthening the argument for the effectiveness of the proposed approach.

5. Selection of Few Labeled Data and Sensitivity to Distribution Gaps:
The method's sensitivity to the selection of the few labeled data and potential distribution gaps between the labeled and unlabeled datasets is a critical concern that warrants further exploration. Understanding how the selection process and any distribution disparities affect the performance of the method is essential for assessing its robustness and applicability to real-world scenarios. Addressing these concerns could greatly enhance the paper's contributions by providing guidelines for effectively implementing the proposed method in diverse settings.

[1] Large language models are human-level prompt engineers

[2] Joint Prompt Optimization of Stacked LLMs using Variational Inference

---

> ### Author Response · Authors · 2024-04-06
> **Response to Reviewer W2o1 (Part 1/2)**
>
> We thank Reviewer W2o1 for finding our work an innovative and timely topic. Below are our responses to the specific concerns.
>
> **Inaccuracy in Abstract regarding Prompt Learning:**
>
> Thank you for the suggestion. To provide a more holistic view of prompt learning, firstly, we have modified the statement of the abstract to remove the explicit mention of *continous* soft prompts. Also, we have included discrete prompt learning methods [1, 2] in the paper's literature to acknowledge the broader research landscape.
>
> **Lack of Distinction between Equations 4 and 5:**
>
> Thanks for pointing it out. Indeed, both equation 4 and 5 have the same mathematical formulation, only differing in the subscript of primary/auxiliary networks. However, it must be noted that there is a key difference in the computations of $x^p$, $w^p$ with that of $x^a$, $w^a$. As seen from Figure 2 of the main paper, ($x^p$, $w^p$) and ($x^a$, $w^a$) are functions of ($T_p$, $V_p$) and ($T_a$, $V_a$) respectively, which have different lengths of prompts associated in the two different pathways. $T_p$, $V_p$ uses a prompt length of $N$ while $T_a$, $V_a$ uses a prompt length of $N/2$. Keeping the same formulation, we can club both the equations as:
>
>  $p(y^j_{c_i}|\mathbf{I}_i) = \frac{\exp(sim(\mathbf{x}_{l,i}^j,\mathbf{w}^j_{c_i})/\tau)}{\sum_{c=1}^{C}\exp(sim(\mathbf{x}_{l,i}^j,\mathbf{w}^j_c)/\tau)}$, where $j \in \{p,a\}$
>
> However, this would seem to clutter the concept, making it more complex in terms of the superscript to interpret. Also, it does not appropriately bring out the implicit difference between the ($x^p$, $w^p$) and ($x^a$, $w^a$) having different prompt length associations.
>
> **Questioning the Advantage of the “Networks” Column in Table 1:**
>
> We thank the reviewer of highlighting this fact. Indeed, efficiency is a critical consideration in the deployment of the proposed model. We would like to clarify that during deployment, only the primary network used. We have mentioned it in `Inference` of Section 3.2 as well as in the caption of the main figure in Figure 2. So no additional network overhead is incurred during the deployment of the model. The auxilliary pathway is used only during training to aid the primary pathways. For training, we allow a trade-off between accuracy and efficiency to get significantly high performance gains without incurring additional computation during deployment.

---

> > ### Author Response · Authors · 2024-04-06
> > **Response to Reviewer W2o1 (Part 2/2)**
> >
> > **Exclusion of Baseline Comparisons in Sections 4.2 and 4.3:**
> >
> > Thanks for the additional baseline suggestion. It will be useful to add baseline comparison that lacks any visual/text prompt. To do so, we ran additional experiments using the CLIP[3] approach on all 15 datasets and included the results as a baseline for comparison with XPL. As observed in the table below, our co-teaching approach exploits multiple prompt learners to learn richer representations using complimentary knowledge and thus outperform with a high margin.
> >
> > Dataset|CLIP|XPL 1-shot|XPL 2-shot|XPL 1%| XPL 5%| XPL 10%|
> > | -------- | -------- | -------- | -------- |-------- |-------- | -------- |
> > EuroSAT|47.8|80.1|81.9|93.8|95.8|97.6|
> > ISIC|12.1|57|57.4|71.5|78.8|85.7|
> > ChestX|13.7|17.2|24|34.2|36.2|36.9|
> > CropDisease|25.6|73.1|87.5|96.5|98.4|98.7|
> > ImageNet|66.0|66.1|67.2|66.2|68.2|72.9|
> > DeepWeeds|18.6|27.6|39.1|76.3|89.1|89.9|
> > Kaokore|14.6|21.7|23.3|35.5|42.2|43.9|
> > DTD|43.8|53.8|59.3|53.8|62|68.7|
> > USPS|49.6|65.6|77.9|70|86.3|90.2|
> > Clevr-Count|19.7|20.2|22.8|56.7|78.2|79.8|
> > UCF-101|67.0|69.4|74.1|68.2|79.5|84.6|
> > Caltech101|93.0|92.9|94.2|92.7|96.3|96.2|
> > OxfordPets|89.1|87.4|88.8|89.4|92.7|93.7|
> > StandfordCars|65.3|65.1|71.0|66.4|77.8|86.5|
> > Flowers102|71.3|87.7|92.5|89.2|94.9|97.6|
> >
> > **Selection of Few Labeled Data and Sensitivity to Distribution Gaps:**
> >
> > Thanks for the reviewers interest in XPL's capability for handling domain shifts. We explain the results of Table 3 included in `Robustness to Domain Shift in Unlabeled Data` (refer Section 4.2 of main paper) in a greater detail. Considering the two domains $\mathcal{D}$ and $\hat{\mathcal{D}}$, the labeled data is always selected from $\mathcal{D}$ itself. The extent of distribution gaps between the labeled and unlabeled domains is controlled by the  $\eta$ hyperparameter, which specifies the amout of unlabeled data to be selected between $\mathcal{D}$ and $\hat{\mathcal{D}}$ having different distributions. When $\eta=0$, we allow no distribution gap as both the labeled and unlabeled data are selected from the same domain $\mathcal{D}$. However, when $\eta=1$, the distribution shift is maximum as the unlabeled samples are entirely selected from $\hat{\mathcal{D}}$. The improvement is comparatively less, but it generalizes well to bridge the domain gap between the labeled and unlabeled data. As observed, our `XPL` excels in handling semi-supervised prompt learning both at the same and different distribution shifts, highlighting the generalizability of our approach.
> >
> > [1] Large language models are human-level prompt engineers
> >
> > [2] Joint Prompt Optimization of Stacked LLMs using Variational Inference
> >
> > [3] Learning transferable visual models from natural language supervision. PMLR, 2021.

---

### Review · Reviewer_tbUN · 2024-03-25

**Summary Of Contributions:**

This paper addresses the problem of adjusting pre-trained vision-language models to multiple downstream tasks. Instead of fine-tuning the VLM's weights, the paper focuses on the prompts being used as input to the VLM. Adjusting prompts for each new downstream task requires domain-specific heuristics, and this paper aims to alleviate that. The approach works by generating multiple views of unlabeled data and incurring a loss in learning similar representations between the views. The experimental results show improvements on a variety of 16 datasets.

**Audience:**

Yes

**Broader Impact Concerns:**

Semi-supervised learning has a positive impact, both on being able to use more data, and possibly to save compute.

**Claims And Evidence:**

Yes

**Requested Changes:**

Question:

* Figures 1c and 1d show that the proposed XPL approach performs similarly to the CoOp approach at only a 1% labeling rate. However, what are the absolute numbers of data? It is not clear if either a) the amount of labeled data is fixed, and this method manages to use ‘additional’ unlabelled data, or b) the amount of total data is fixed, and this method manages to make do with ‘less’ labeled data. (Same for Figure 3)
* The semi-supervised approach seems similar to ‘Noisy-student’ from [3]. What are the similarities/differences w.r.t. that paper?

**Strengths And Weaknesses:**

Strengths:

* This work makes a bridge between prompt learning and semi-supervised learning.
* The paper has multiple experimental results to assess various aspects of the improvements.
* Various settings for prompt learning are compared with, c.f. Table 1.


Weaknesses:

* Although the paper shows promising experimental results, a more theoretical or grounded explanation for the method is missing. One could interpret the method of ‘augmenting multiple views and regressing a common representation’ as maximizing/minimizing specific forms of information [1][2]. Such an explanation could help to interpret Figure 2.
* Most datasets in Section 4.1 are relatively small, other than ImageNet. Although the paper argues for few-show learning, that performance could still be measured on larger datasets.

[1] Oord, Aaron van den, Yazhe Li, and Oriol Vinyals. "Representation learning with contrastive predictive coding." arXiv preprint arXiv:1807.03748 (2018).

[2] Michael Tschannen, Josip Djolonga, Paul K. Rubenstein, Sylvain Gelly, Mario Lucic:
On Mutual Information Maximization for Representation Learning. ICLR 2020

[3] Xie, Qizhe, et al. "Self-training with noisy student improves imagenet classification." CVPR. 2020.

---

> ### Author Response · Authors · 2024-04-06
> **Response to Reviewer tbUN (Part 1/2)**
>
> We thank Reviewer tbUN for acknowledging that our approach bridges prompt learning and semi-supervised learning. Below are our responses to the specific concerns.
>
> **Grounded explanation for the method, as one could interpret the method of ‘augmenting multiple views and regressing a common representation’ as maximizing/minimizing specific forms of information [1][2]:**
>
> In our proposed approach, we introduce different length prompts as a key component in generating mutual information. For a more grounded analysis of the complimentary training using multiple prompt learners, we evaluate the classwise performance between two models leveraging same amount of labeled and unlabeled data but with different number of learnable prompts. As shown in Figure 1a and 1b of the main paper, we observe that two models leveraging unlabeled data but with different number of learnable prompts exhibit markedly different category-wise performance. This is further analyzed in the Ablation Studies (refer Section 4.3), where we try a variation of different prompt length combinations and also relative prompt positions to analyze how the performance varies with the mutual information derived from each combination and how well they compliment each other. Further, we analyze the latent representations between the uni-modal and multi-modal prompt learning baselines in terms of learning discriminative features by feature visualizations using t-SNE (refer A.6 of appendix). Here we observe that multi-modal prompts of different lengths contribute towards more consistent clustering and class-wise discriminative performance.
>
> *Performance of XPL on other larger datasets:**
>
> Thanks for the suggestion. Besides the results shown in the main paper over 16 datasets, to acknowledge the reviewers interest for XPL's performance over a relatively larger dataset, we ran additional experiments on the SUN397 dataset. It has 397 categories and is much larger than most of the datasets already used in the paper. Given the time constraint, we evaluate XPL along with the next best baseline of MPL$^u$ and the uni-modal baseline of TPL on this dataset for 1-shot, 2-shot, 1% and 5% labeled data setting. The results are provided in the table below. As observed, even for the larger dataset, XPL continues to maintain superior performance over the other baslines.
>
>  | SUN397 | 1-shot | 2-shot | 1% | 5% |
>  |-------- | -------- | -------- | -------- | -------- |
> |XPL| 68.3 | 69.8 | 68.3 | 71.7
> |MPL$^u$ |67.6 | 69.0 | 67.4|70.3
> |TPL|66.6|67.0|66.2|67.4
>
> **Figures 1c and 1d show that the proposed XPL approach performs similarly to the CoOp approach at only a 1% labeling rate. However, what are the absolute numbers of data? It is not clear if either a) the amount of labeled data is fixed, and this method manages to use ‘additional’ unlabeled data, or b) the amount of total data is fixed, and this method manages to make do with ‘less’ labeled data:**
>
> Referring to Figures 1c of the main paper for EuroSAT dataset, the red dotted line depicts CoOp-100%. This refers to running the CoOp approach under full supervision using all 13,500 images of the training set as labeled data. The accuracy for CoOp-100% in EuroSAT is 94.2%. Considering our proposed XPL, we give the accuracy values for 1%, 5% and 10% labeled data considering only 135, 675 and 1350 images out of the 13,500 images of the training set as labeled data respectively. Remaining 13365, 12825 and 12150 images are used as the unlabeled set for the 1%, 5% and 10% cases, respectively. The corresponding accuracy values of XPL for 1%, 5% and 10% are 93.8%, 95.8% and 97.6%, respectively. So this highlights that, using fixed amount of total data, this method manages to make do with ‘less’ labeled data by leveraging on remaining unlabeled data.
> Similarly, for Figures 1d of the main paper showing the CropDisease dataset, the total number of training images is 43,456. CoOp-100% uses all this data as labeled and achieves 94.5 whereas using only 1%, 5% and 10% of 43,456, XPL achieves superior accuracies of 96.5%, 98.4% and 98.7% respectively.

---

> ### Author Response · Authors · 2024-04-06
> **Response to Reviewer tbUN (Part 2/2)**
>
> **Similarities/differences w.r.t. ‘Noisy-student’ from [1]:**
>
> Our proposed XPL and ‘Noisy-student’ from [1] are both semi-supervised approaches bearing some similarity but it has distintive differences seperating their respective application scopes.
>
> *Similarites:*
>
> 1. Both leverage on larger corpus of unlabled images to learn richer representations
> 2. Both approaches use a two pathway framework for processing unlabeled samples
> 3. Both training policies rely on generating high quality pseudo labels for the unlabeled samples
>
> *Differences:*
>
> 1. Our proposed XPL is a multi-modal approach including both text and image modalities where training loss in minimized on a combined embedding space for the two modalities. Noisy-student on the other hand is a uni-modal approach only using image features.
> 2. One of the key goals of XPL is parameter-efficient finetunig of VMLs, learning only a handful of soft prompts. Whereas, Noisy-student trains the entire EfficientNet model updating all its model parameters, requring much higher compute complexity.
> 3. Although Noisy-student has a two pathway framework, it relies on a teacher-student training policy, that to, updating one network in every interation. They train the teacher first in one iteration and then generate pseudo labels to train the student in the next iteration. This is distinctly different from XPL. Firstly, rather than a teacher-student approach, XPL relies on a cross-model complimentary training strategy where each pathway with different number of learnable prompts learns different category-wise represenataions. Both learn complimentary knowledge and thus provide semi-supervision to each other, updating the leanable prompts in both the pathways in same iteration.
> 4. In Noisy-student the size of student network is also increased. They take a larger EfficientNet backbone compared to the teacher model, as a student model for semi-supervised learning on the combination of labeled and teacher generated pseudo labeled images. This further increases the computation. However in XPL, the frozen VLM encoder backbones are fixed. The only difference in the two pathways is in the lengths of prompt vectors assicociated in each pathways.
> 5. In Noisy-student the teacher model is not noised to favour higher quality of pseudo labels for the student. The noise such as dropout and data augmentation via RandAugment are only applied on the student. In our case, as XPL uses a cross-model complimentary training, both the pathways are fed with weakly and strongly augmentationed samples thereby making the complimentary knowledge enrich the pseudo labels across both the pathways simultaneously.
> 6. Lastly, the semi-supervision of Noisy-student doesnot include the invaricance towards prompt learning. XPL is the first work which introduces prompt inviariance as a component of cosistency regularization in the semi-supervision learning policy.
>
> [1] Xie, Qizhe, et al. "Self-training with noisy student improves imagenet classification." CVPR. 2020.

---

### Review · Reviewer_LUr8 · 2024-03-27

**Summary Of Contributions:**

The paper explores the effect of prompt learning from unlabeled data for pretrained vision-language models (VLMs), especially in low-labeled data regime (few-shot learning or semi-supervised learning). With both supervised multi-modal training on labeled data and unsupervised multi-modal, cross-model training on unlabeled data, the proposed method, XPL, achieves superior results on 15 diverse image classification datasets.

**Audience:**

Yes

**Claims And Evidence:**

Yes

**Requested Changes:**

Please address the concerns raised above.

**Strengths And Weaknesses:**

Strengths
- Overall, the paper is well-written and easy to read.
- The paper addresses and validates the concept of training on unlabeled data for prompt learning in low-labeled data regime. With the cross-model and multi-modal prompt learning, the proposed method achieves superior results in comprehensive experiments. The ablation studies also validate the design choices the authors made, which strengthens quality of the paper.


Weaknesses
- What are the main contributions the authors claim? Unsupervised training for prompt learning (MPL vs MPL^u), cross-model prompt learning with different number of prompt vectors (MPL^u vs XPL), multi-modal prompt learning (TPL/VPL vs MPL), or all of them? Which are the ones that this paper first try or there are previous works exploring similar approaches to? Please make this clear.
- All the models used in the experiments are based on the same architecture and training objectives with the addition/subtraction of some components. Regarding the above point, if there are previous works that explored similar approaches, the authors should compare with them.
- While the authors provide extensive experiment results to validate the proposed method, there are still remaining design choices not addressed. For example, does XPL use any constraints to stabilize cross-model representation learning, such as stop gradients as in BYOL [1]? Why did the authors use hard (one-hot) pseudo-labels for unsupervised training? Why did the authors use only weak augmentation for supervised training? And does scaling up model capacity (size) affect the generalization performance of XPL? (Figure 5 compared the same size models with different training recipes). The authors also did not explicitly show the effectiveness of learning prompts invariant to different views of the same image, which needs comparison with the cross-model training, but using weak augmentation only.
- Some experimental details are missing, e.g., how the authors split the classes of each dataset for evaluating generation from seen to unseen classes (Table 2), and how they sampled 1%/5%/10% of labeled data.
- MoCo is a self-supervised baseline for learning visual (that is, image-only) representation while XPL uses cross-modal contrastive losses, so the comparison in figure 6 seems to be not fair to me. Please elaborate on how the authors applied MoCo for the experiment.

Minor questions
- Appendix A.1 does not include comparison between MPL and TPL/VPL.

[1] Jean-Bastien Grill et al., Bootstrap your own latent: A new approach to self-supervised Learning. NeurIPS 2020

---

> ### Author Response · Authors · 2024-04-06
> **Response to Reviewer LUr8 (Part 1/2)**
>
> We thank Reviewer LUr8 for finding our paper well-written and easy to read. Below are our responses to the specific concerns.
>
> **What are the main contributions the authors claim and if there are previous works that explored similar approaches?**
>
> We thank the reviewer for this interesting query regarding the different contributions of our paper. One of the key contributions of our work is to go beyond the scope of conventional supervised prompt learning and leverage on the unlabeled samples achieving significant gain in model performance. For this, we explore three different components, namely, leveraging unlabeled data, multiple prompt learners and multi-modal prompts. As this work is one of the *first* works in multi-modal semi-supervised prompt learning, we carefully design multiple baselines for assessing these components and their combinations. Firstly, we show in (MPL vs MPL$^u$) how even naive approach to harness unlabeled data for prompt learning can add to the performance gain. This enables us to use the information from unlabeled data. Next in (MPL$^u$ vs XPL), we utilize a cross-model framework to learn complimentary knowledge from multiple prompt learners to boost the performance. This also falls in line with our observations that two models leveraging unlabeled data but with different number of learnable prompts exhibit markedly different category-wise performance (ref. Figure 1a and b of main paper). Lastly, in (TPL/VPL vs MPL) we justify the use of multi-modal prompts, including both text and visual prompts. This shows why multi-modal prompt learning is more beneficial and generates richer representations. These baseline comparisons contribute to an inclusive and comprehensive assessment of different combinations of semi-supervised and prompt learning strategies.
>
>
> **Does XPL use any constraints to stabilize cross-model representation learning, such as stop gradients as in BYOL?**
>
> Thanks for the suggestion. In BYOL, the target network provides the regression targets and functions as a teacher to make the training of the online network more through. The target network is in turn updated the with a slow-moving average of the online network. This boostrapping startegy requires stabilization which they achieve by stop gradient for the online network. However, our proposed approach relies more on complementary training rather than a teacher-student approach. Both the networks contribute equally to learn corresponding category-wise representations and thus complement in providing semi-supervision to each other. So in our case, rather than bootstapping stabilization, we implement consistency regularization by achieving model invariance towards data perturbations as well as prompt length variations.
>
> **Why did the authors use hard (one-hot) pseudo-labels for unsupervised training?**
>
> Pseudo labeling forms one of the fundamental approaches of unsupervised training[1]. For XPL, given an unlabeled image, we use the hard (one-hot) pseudo-labels to bring confident prediction from the auxiliary network to train the primary network and vice versa. We minimize the cross-entropy loss between pseudo-labels generated by the auxiliary network and the predictions made by the primary and vice versa. The use of a hard label makes pseudo-labeling closely related to entropy minimization, where the model’s predictions are encouraged to be low-entropy (i.e., high-confidence) on unlabeled data.
>
> **Why did the authors use only weak augmentation for supervised training?**
>
> Supervised training relies on the groundtruth classes for minimizing the cross-entropy loss for the predictions of the primary network. The main purpose of using both weak and strong augmentations is to create different versions of a given unlabeled sample and force the model to learn invariant representations. Training models to learn such representations is necessary in absense of label annonations to reduce the entropy. When the ground-truth labels are itself present, introducing an additional set of versions using strong augmentations would significantly increase the compute cost and also undermine the entropy minimization obtained directly using the ground-truths.

---

> ### Author Response · Authors · 2024-04-06
> **Response to Reviewer LUr8 (Part 2/2)**
>
> **Does scaling up model capacity (size) affect the generalization performance of XPL?**
>
> We evalute XPL using different VLM backbones of varying model capacities. We show the performance of XPL on ViT-B/16 (refer Figure 3 of main paper),  ViT-B/32 and DeCLIP w/ ViT-B/32 (refer Figure 5 of main paper). Comparing the average accuracy plots across 15 datasets, we observe that XPL maintains superiority across all model-scales over the considered baselines. However, comparing on the XPL performances, it is observed that the performance maynot always translate proportionally with the scaling up of model capacity. The intution being the length of prompts being associated. As the model capacity is scaled up, using same set of lower prompt lengths may lead to a performance drop because of the very low number of learnable parameters. In that case higher prompt lengths need to be used to cope with larger models.
>
> **Effectiveness of learning prompts invariant to different views of the same image, which needs comparison with the cross-model training, but using weak augmentation only:**
>
> To generate two different views of the same image, we need two different sets of augmentations, one weak and other strong. Using only weak augmentations would generate a single view of the data. This would restrain the consistency regularization on the unlabeled samples. In our complementary training, the weakly augmented versions generate the pseudo-labels for minimizing the loss against the strongly augmented sample predictions from the auxiliary network and vice versa.
>
> **More experimental details on generalization experiments:**
>
> Thanks for the query. For the generalization experiments from Seen to Unseen Classes, given a dataset, we split the classes equally into two groups, one group is considered as the `Seen` classes and the other as `Unseen` classes. For selecting the proportion of labeled data (1%/5%/10%), we randomly choose the certain percentage of labeled samples from each class and discard the labels for the remaining data to form a large unlabeled set.
>
> **Elaborating how MoCo was applied for the experiment:**
>
> For applying MoCo in XPL, we replaced the cross-model consistency regularization with a single network with additional momentum encoder. Also, instead of providing a weakly and strongly augmented version of a given image, we followed the MoCo's policy to input a single image as query and its augmented image as a key. Only in our multi-modal scenario, the contrastive loss is applied on the prediction probabilities dervied from the joint embedding space as computed from Equations 6 and 7 of the main paper. The difference being that the visual embeddings x$^{p,wk}$ and x$^{p,str}$ are replaced with x$^{p,query}$ and x$^{p,key}$ derived from the query and key respectively.
>
> **Why Appendix A.1 does not include comparison between MPL and TPL/VPL?**
>
> Appendix A.1 includes the section of `Leveraging Unlabeled Data for Uni-modal baselines` where we only showcase the uni-modal baselines of TPL, TPL$^u$, VPL and VPL$^u$. So, MPL being a multi-modal baseline including both text and visual prompts has been excluded from this section. The comparison between MPL and TPL/VPL is portrayed in the Figure 3 of the main paper.
>
> [1] Fixmatch: Simplifying semi-supervised learning with consistency and confidence, Advances in Neural Information Processing systems 2020

---

> > ### Comment · Reviewer_LUr8 · 2024-04-10
> >
> > Thank you for the responses.
> > Regarding my minor question, the authors mentioned Appendix A.1 for comparison between MPL and TPL/VPL in 1-shot classification, in Section 4.2 "Multimodal Prompt Learning".

---

> > > ### Author Response · Authors · 2024-04-10
> > > **Response to Reviewer LUr8 feedback**
> > >
> > > Thank you for the feedback and we’re glad that our response addressed all your concerns.

---

### Author Response · Authors · 2024-04-06
**Summary of Author's Response**

We would like to thank all the reviewers for their constructive comments! We are encouraged that the reviewers appreciate our innovative approach in bridging the concepts of prompt learning and semi-supervised learning (Reviewers tbUN, W2o1, G6Rs). We are glad that the reviewers find (a) our paper well-written and easy to read (Reviewer LUr8) and (b) our experiments and ablation studies to be comprehensive and thorough (Reviewer tbUN, W2o1, G6Rs).
We have addressed all the questions that the reviewers posed with additional experimental comparisons and clarifications.

---

### Decision · Action_Editor_NRWv · 2024-05-19

**Recommendation:** Accept with minor revision

**Comment:**

The paper proposes a novel framework, XPL, for semi-supervised prompt learning in vision-language models (VLMs). This approach leverages unlabeled data to enhance the adaptability of pretrained VLMs to multiple downstream tasks without extensive retraining. XPL employs a dual model architecture where different prompt lengths learn from unlabeled samples through a cross-model design, improving learning efficiency and generalization.

There is a unanimous decision to accept the paper. Using semi-supervised learning on prompt learning for multimodal models is a novel contribution. The methodology is rigorously tested on diverse datasets with significant improvements. The framework also shows strong performance with out-of-domain unlabeled images. On the negative side, there are concerns regarding: 1) the lack of theoretical grounding for why the method works, 2) connections to existing literature in both prompt tuning and semi-supervised learning, and 3) other experimental improvements and clarifications. The authors did a good job addressing the concerns with their rebuttal. Overall, this is a solid work with an interesting problem, comprehensive results, and clear improvements. The AE thus recommends acceptance to TMLR. The authors should carefully revise their paper and address all reviewer concerns in their final version.

**Audience:**

The topic is of considerable interest to the audience of the multimodal, semi-supervised learning and general transfer learning communities.

**Claims And Evidence:**

All reviewers affirmed that the paper’s claims are substantiated by clear and convincing evidence. The innovative use of semi-supervised learning to improve prompt learning in vision-language models through the XPL framework has been validated across multiple datasets.